# Hydrogen peroxide signaling via its transformation to a stereospecific alkyl hydroperoxide that escapes reductive inactivation

Raphael F. Queiroz [1,2], Christopher P. Stanley[2,3], Kathryn Wolhuter[2], Stephanie M. Y. Kong [3], Ragul Rajivan [2], Naomi McKinnon[2], Giang T. H. Nguyen [4], Antonella Roveri [5], Sebastian Guttzeit [6], Philip Eaton [7], William A. Donald [4], Fulvio Ursini[5], Christine C. Winterbourn [8], Anita Ayer [2,3,9 ✉] & Roland Stocker [2,3,9,10 ✉]

During systemic inflammation, indoleamine 2,3-dioxygenase 1 (IDO1) becomes expressed in endothelial cells where it uses hydrogen peroxide ($H_2O_2$) to oxidize L-tryptophan to the tricyclic hydroperoxide, *cis*-WOOH, that then relaxes arteries via oxidation of protein kinase G 1α. Here we show that arterial glutathione peroxidases and peroxiredoxins that rapidly eliminate $H_2O_2$, have little impact on relaxation of IDO1-expressing arteries, and that purified IDO1 forms *cis*-WOOH in the presence of peroxiredoxin 2. *cis*-WOOH oxidizes protein thiols in a selective and stereospecific manner. Compared with its epimer *trans*-WOOH and $H_2O_2$, *cis*-WOOH reacts slower with the major arterial forms of glutathione peroxidases and peroxiredoxins while it reacts more readily with its target, protein kinase G 1α. Our results indicate a paradigm of redox signaling by $H_2O_2$ via its enzymatic conversion to an amino acid-derived hydroperoxide that 'escapes' effective reductive inactivation to engage in selective oxidative activation of key target proteins.

[1] Department of Natural Sciences, Southwest Bahia State University, Vitoria da Conquista, Bahia, Brazil. [2] Victor Chang Cardiac Research Institute, Darlinghurst, New South Wales, Australia. [3] Heart Research Institute, The University of Sydney, Sydney, Australia. [4] School of Chemistry, University of New South Wales, Sydney, New South Wales, Australia. [5] Department of Molecular Medicine, University of Padova, Padova, Italy. [6] King's College, The Rayne Institute, St Thomas' Hospital, London, UK. [7] William Harvey Research Institute, Barts and the London School of Medicine, Queen Mary University of London, London, UK. [8] Centre for Free Radical Research, Department of Pathology, University of Otago Christchurch, Christchurch, New Zealand. [9] St Vincent's Clinical School, University of New South Wales, Sydney, New South Wales, Australia. [10] School of Life and Environmental Sciences, The University of Sydney, Sydney, Australia. ✉email: anita.ayer@hri.org.au; roland.stocker@hri.org.au

Research on redox signaling involving 2-electron reactions commonly focuses on $H_2O_2$ as the oxidant messenger. At present, four models have been proposed to explain $H_2O_2$-mediated redox signaling: the "floodgate" hypothesis, transient inactivation, redox relay, and direct activation model. In the floodgate hypothesis, high concentrations of $H_2O_2$ are proposed to hyperoxidize reducing enzymes, mainly peroxiredoxins (Prxs), leading to a temporary local inactivation of the peroxidases and buildup of $H_2O_2$ to concentrations sufficient to oxidize a Cys residue in a particular signaling protein[1–3]. Similarly, the transient inactivation model hypothesizes a transient inactivation of Prx via phosphorylation[4]. The redox relay model proposes Prx to initially react with $H_2O_2$ and oxidized Prx to then transmit the oxidizing equivalents to the signaling protein via formation of a heterodimeric protein disulfide complex[5,6]. Last, the direct activation model assumes $H_2O_2$ to react faster with its target signaling protein than Prx and glutathione peroxidases (GPx). A significant challenge for all models, especially the direct oxidation model is the fact that $H_2O_2$ itself is a strong, but kinetically-hindered oxidant, and the rate of its reaction with target protein thiol groups is kinetically too slow ($\sim 10^1\,M^{-1}\,s^{-1}$) to outcompete scavenging of $H_2O_2$ by Prx and GPx, except perhaps in redox stress conditions[7,8].

We recently reported a mechanism of $H_2O_2$-mediated redox signaling in vivo[9]. Under inflammatory conditions, heme-containing indoleamine 2,3-dioxygenase 1 (IDO1) becomes expressed in arteries[10] where it requires $H_2O_2$ to form singlet oxygen that, in turn, oxidizes L-Trp into the tricyclic hydroperoxide cis-WOOH[9] (Fig. 1a). cis-WOOH causes oxidative dimerization of Cys42 residue of protein kinase G 1α (PKG1α), and this results in relaxation of resistance arteries and an associated decrease in blood pressure. In contrast, the epimer trans-WOOH (Fig. 1a) is inefficient in causing oxidative dimerization of PKG1α and it does not promote arterial relaxation[9]. Also, while oxidation of L-Trp by reagent singlet oxygen or myeloperoxidase plus $H_2O_2$ gives rise to similar ratio of cis- and trans-WOOH, only cis-WOOH is formed by IDO1 plus $H_2O_2$[9]. These findings highlight the role of cis-WOOH as a stereoselective redox signaling molecule in arteries in inflammation. Nevertheless, and similar to the situation with $H_2O_2$[7,8], a key question that remains to be elucidated is how cis-WOOH can 'redox signal' in the presence of enzymatic systems designed to rapidly remove hydroperoxides.

As implied above, the two major thiol-containing reducing systems for the removal of hydroperoxides and maintenance of cellular protein thiol redox state are linked to thioredoxin and glutathione (GSH). The former is composed of Prx, thioredoxin, and thioredoxin reductase (TrxR), while the GSH reducing system consists of GPx, GSH, and glutathione reductase (GR)[11,12]. Both depend on NADPH as the ultimate reductant[13–15]. GPx4 and Prx1-4 also reduce alkyl hydroperoxides. GPx4 reduces phosphatidylcholine hydroperoxide[16] with a second-order rate constant of $4 \times 10^7\,M^{-1}\,s^{-1}$[17]. In the case of Prx1-4, the rate constants for their reaction with alkyl hydroperoxides range widely (i.e., $10^2$–$10^6\,M^{-1}\,s^{-1}$)[18–20] and generally decrease with increasing bulkiness of the substrate[18,19]. Available literature indicates that Prx2 reacts relatively slowly with amino acid-derived hydroperoxides, with the highest second-order rate constant reported for the reaction of N-acetyl-leucine hydroperoxide and Prx2 being $4 \times 10^4\,M^{-1}\,s^{-1}$[19], i.e., about 3 orders of magnitude lower than that for the reaction of Prx2 with $H_2O_2$[21]. To date, the rates of reaction of GPx4 and Prxs with cis-WOOH have not been determined.

Information on the distribution of various GPx and Prx isoforms in different arterial beds and how these enzymes impact arterial redox signaling is also limited. Eaton and colleagues reported the content of Prxs (not specific to Prx isoforms) to be higher in mouse aortas than mesenteric arteries and suggested that this may help explain the comparatively lower sensitivity of the aorta to $H_2O_2$-induced relaxation compared with mesenteric arteries[22]. Auranofin-mediated inhibition of TrxR and, consequently, Prx recycling, increased oxidative dimerization of PKG1α in mouse aortic rings, likely via $H_2O_2$[22,23]. Similarly, depletion of thioredoxin and TrxR1 by small inhibitory RNA enhanced PKG1α dimerization and relaxation of bovine pulmonary arteries in the presence of $H_2O_2$[24]. These results suggest that Prx/GPx regulate arterial redox signaling by $H_2O_2$ although the impact of inflammation, when endogenous $H_2O_2$ is utilized to form cis-WOOH, on this process remains unclear.

Here we evaluated the presence of different isoforms of GPx and Prx in conduit and resistance mouse arteries and determined the rate of reduction of cis-WOOH by the major peroxidases detected in resistance arteries. We describe stereospecific differences between cis-WOOH and its epimer trans-WOOH in the interaction and reaction with peroxidases that indicate that cis-WOOH behaves in a distinct manner to coordinate redox signaling unlike any other hydroperoxide described. Based on our findings, we propose a model of redox signaling by $H_2O_2$ that depends on its enzymatic utilization to form an amino acid-derived hydroperoxide that then signals in a stereospecific manner.

## Results

**Role of thiol peroxidases in the regulation of arterial tone in mice.** $H_2O_2$ is proposed to physiologically antagonize myogenic constriction via oxidation of PKG1α[25]. We therefore hypothesized that $H_2O_2$ may act in a similar manner opposing noradrenaline-induced constriction (Fig. 1b), and that such effects are subject to regulation by Prx and GPx that metabolize $H_2O_2$ (Fig. 1c). To block recycling of oxidized Prx, we inhibited thioredoxin reductase using auranofin (AUR) (Fig. 1c)[26] in noradrenaline pre-constricted mesenteric arteries, hypothesizing that this would increase $H_2O_2$ concentrations and hence $H_2O_2$-mediated arterial relaxation (Fig. 1c). Indeed, addition of AUR (300 nM) caused relaxation (Fig. 1d) in a concentration-dependent manner (Supplementary Fig. 1a). AUR relaxed arteries pre-constricted with the depolarizing agent potassium chloride only modestly (Supplementary Fig. 1b), consistent with $K^+$ channels being downstream of $H_2O_2$-dependent relaxation[25]. AUR-induced relaxation was comparable in the presence and absence of L-NAME and indomethacin (Supplementary Fig. 1c). This suggests that $H_2O_2$-dependent arterial relaxation occurred in the presence of active endothelial nitric oxide synthase and cyclooxygenase, consistent with the in vivo data showing peroxide-mediated arterial relaxation in the presence of active vasodilatory pathways[9].

To block recycling of oxidized GPx, we inhibited glutathione reductase in noradrenaline pre-constricted arteries using [1,3-bis(2-chloroethyl)-1-nitrosourea] (BCNU)[27] (Fig. 1c). BCNU relaxed pre-constricted mouse mesenteric arteries in a dose-dependent manner (Fig. 1d and Supplementary Fig. 1d), and this effect was enhanced when BCNU was added together with AUR (Fig. 1e). To confirm that AUR- and BCNU-induced relaxation of resistance arteries was associated with oxidation of PKG1α, and hence consistent with endogenously formed $H_2O_2$[28], we investigated responses using arteries from wild type and PKG1α C42S knock-in "redox dead" mice. Compared with wild-type controls, AUR and BCNU-induced relaxation was significantly attenuated in pre-constricted resistance arteries from redox dead mice (Fig. 1f) consistent with relaxation being mediated by $H_2O_2$. Under systemic inflammatory conditions, arteries express IDO1[9,10] and form cis-WOOH[9]. To determine the role of GPx

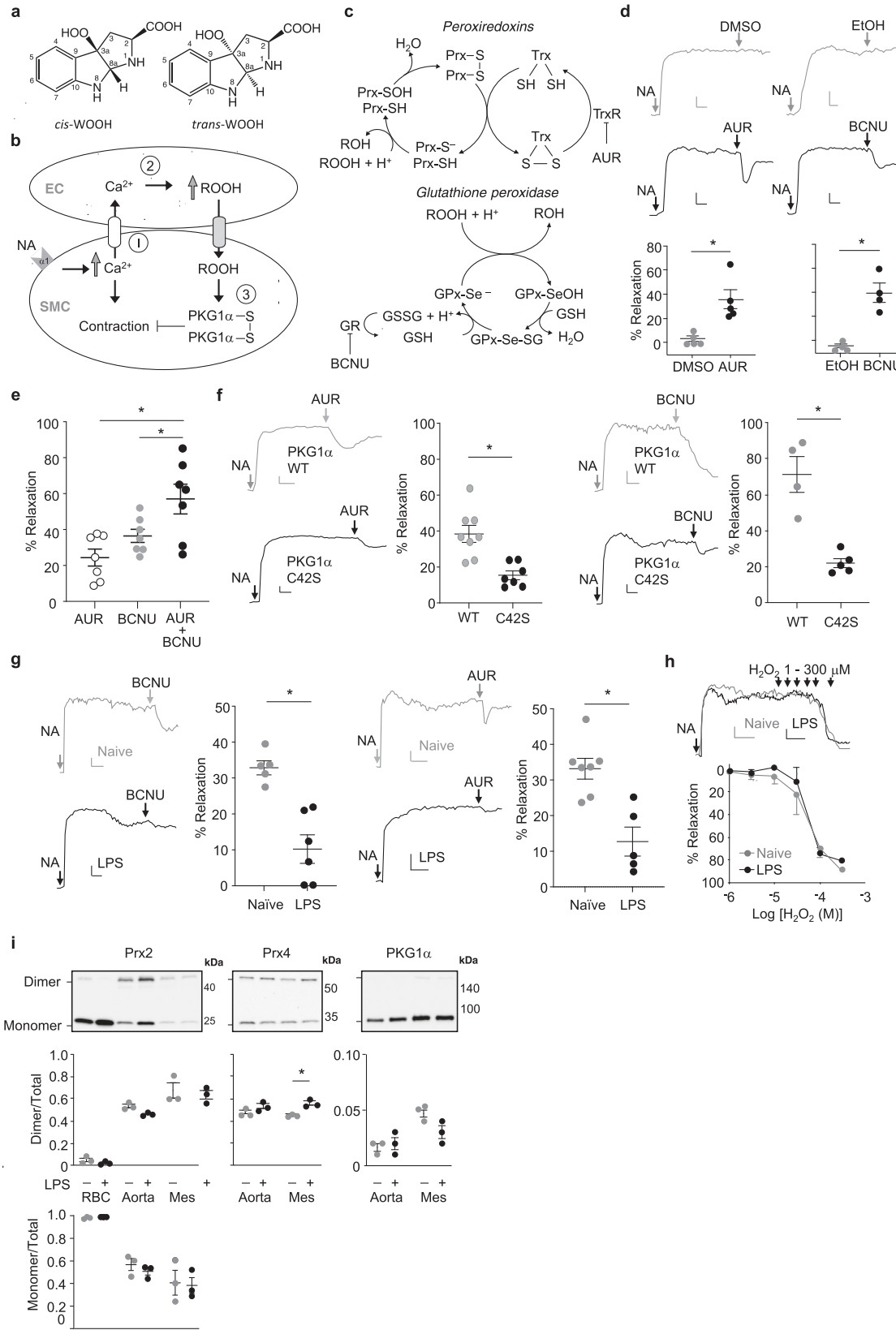

and Prx under these conditions, mesenteric arteries from mice treated with lipopolysaccharide (LPS) for 16 h were exposed to BCNU or AUR. Arterial relaxation was significantly attenuated under these conditions (Fig. 1g). This was not caused by an LPS-induced overall decline in sensitivity to bolus (1–300 μM) $H_2O_2$-induced arterial relaxation (Fig. 1h), indicating that LPS does not

cause substantial changes to either endogenous defense systems against $H_2O_2$ or oxidant-induced activation of PKG1α.

Immunoblotting for Prx and GPx isoforms in abdominal aorta and mesenteric arteries demonstrated GPx4, Prx2, and Prx4 to be the most abundantly expressed peroxidases (Table 1, Supplementary Fig. 2). GPx1 and Prx1 were present at comparatively

**Fig. 1 Role of thiol peroxidases in the regulation of arterial tone in mice. a** Epimers of tryptophan-derived hydroperoxides cis-WOOH and trans-WOOH. **b** Schematic of proposed mechanism by which hydroperoxides antagonize arterial constriction in response to noradrenaline (NA). **c** Schematic representation of the thiol peroxidase redox cycles of peroxiredoxins (Prx), glutathione peroxidases (GPx), and their respective inhibitors auranofin (AUR) and (1,3-bis(2-chloroethyl)-1-nitrosourea, BCNU). **d** Compared with vehicle control (0.01% DMSO or EtOH), AUR (300 nM; left panel) and BCNU (100 µM; right panel) cause arterial relaxation of NA pre-constricted mesenteric resistance arteries from C57BL6/J mice (AUR $n = 5$, BCNU $n = 4$). Top panels show representative traces with respective dot blots shown below. **e** Comparison of relaxation responses from mice treated auranofin (AUR, 300 nM), BCNU (100 µM), and combined AUR and BCNU treatment ($n = 7$). **f** AUR-induced arterial relaxation in C57BL6/N PKG1α C42S mutant ($n = 7$; left panel) and BCNU-induced arterial relaxation in C57BL6/N PKG1α C42S mutant ($n = 5$; right panel) compared with PKG1α WT control mice ($n = 5$–8). **g** BCNU (100 µM; left panel) and AUR (300 nM; right panel) induced relaxation of NA pre-constricted arteries from naïve (BCNU $n = 5$, AUR $n = 7$) and lipopolysaccharide (LPS)-treated mice (BCNU $n = 6$, AUR $n = 5$). Representative traces and respective dot blots shown. **h** Hydrogen peroxide ($H_2O_2$) mediated relaxation of NA pre-constricted mesenteric resistance arteries from naïve ($n = 4$) and LPS-treated ($n = 4$) mice. Top shows representative traces with summary line chart shown below. **i** Redox state of Prx2, Prx4, and PKG1α in aorta and mesenteric arteries (mes) isolated from nave ($n = 3$) and LPS-treated ($n = 3$) mice. The oxidation state of Prx2, Prx4, and PKG1α was captured using N-ethylmaleimide and assessed by quantification of dimer and monomer bands. Red blood cells (RBCs) were used as positive controls for Prx2 dimer/monomer-to-total ratio. Top shows representative blots with respective dot blots shown below. Gray and black indicate control and experimental conditions, respectively. Vascular myography experiments were conducted in the presence of L-NAME (100 µM) and indomethacin (10 µM), with the scale bar depicting 0.5 mN (y-axis) and 2 min (x-axis). Summary data are shown as mean ± SEM, with each individual data point referring to an independent experiment. Statistical analysis was performed using two-tailed Mann–Whitney tests OR a one-way ANOVA with Holm–Sidak multiple comparison test (**e**). *$p \leq 0.05$. Exact p-values are as follows: **d** $p = 0.0286$; **e** $p = 0.0022$ AUR+BCNU vs AUR; $p = 0.0080$, AUR+BCNU vs BCNU; **f** $p = 0.0159$ BCNU panel, $p = 0.0034$ AUR panel, **g** $p = 0.0043$ BCNU panel; 0.0087 AUR panel; **h** $p > 0.9999$, 1 µM; $p = 0.9977$, 3 µM; $p = 0.5198$, 10 µM; $p = 0.2145$, 30 µM; $p = 0.8887$; 100 µM: $p = 0.5125$, 300 µM. Uncropped blots and source data are provided as a Source data file.

**Table 1 Expression of peroxidase enzymes (Prx and GPx) and protein kinase G1α (PKG1α) in abdominal aorta and mesenteric arteries of naïve and LPS-treated mice.**

| | Expression (fmol/µg tissue ± SEM) | | | |
| | Naïve ($n = 3$–6) | | LPS ($n = 3$–5) | |
| Enzyme | Abdominal aorta | Mesenteric artery | Abdominal aorta | Mesenteric artery |
|---|---|---|---|---|
| Prx1 | ND | ND | 3 ± 1* | 3 ± 1* |
| Prx2 | 24 ± 2 | 22 ± 1 | 13 ± 1 | 14 ± 1 |
| Prx3 | ND | ND | ND | ND |
| Prx4 | 27 ± 0* | 13 ± 1*,# | 35 ± 1 | 27 ± 5# |
| Prx5 | ND | ND | ND | ND |
| Prx6 | ND | ND | ND | ND |
| GPx1 | 5 ± 1 | 4 ± 0 | ND | ND |
| GPx4 | 21 ± 7 | 26 ± 5# | 9 ± 1 | 6 ± 2# |
| PKG1α | 7 ± 2 | 7 ± 2 | 8 ± 2 | 7 ± 4 |

Data are shown as mean ± SEM. *$p \leq 0.05$ between abdominal aorta and mesenteric artery under same treatment conditions; and #$p \leq 0.05$ between abdominal aorta and mesenteric artery under different treatment conditions, as determined by two-way ANOVA with Tukey's post hoc test. ND, not detected or below readily quantifiable concentration.

lower concentrations, and Prx3, 5, and 6 were below the detection limit. LPS treatment significantly decreased GPx4 and increased Prx1 and Prx4 in mesenteric arteries whilst LPS had no measurable effect on Prx2 and PKG1α expression (Table 1, Supplementary Fig. 2). To ensure that exposure to LPS did not oxidize and thus inhibit peroxidase activity prior to administering BCNU or AUR negating their effect, the thiol oxidation state of Prx2 and Prx4 along with PKG1α was determined in vivo. To do this, we perfused naïve or LPS-treated mice with phosphate buffered saline (PBS) containing N-ethylmaleimide (NEM) (100 mM) to maintain endogenous thiol status and determined the redox state of Prx2, Prx4, and PKG1α using a non-reducing immunoblot. To control for possible alterations to the thiol redox state during sample work-up, we determined the Prx2 redox state in red blood cells. Red blood cell Prx2 was almost entirely present in its reduced, monomeric form, indicating[28] that inadvertent ex vivo oxidation of Prx2 was minimal under the experimental conditions used. LPS caused overall modest changes in protein thiol oxidation in arteries, with only an increase in mesenteric Prx4 oxidation reaching statistical significance (Fig. 1i). Prx2 and PKG1α expression and their thiol oxidation state were unchanged in response to LPS (Fig. 1i).

**Reduction of cis-WOOH by GPx.** The above findings suggest that GPx and Prx have a decreased ability to inhibit arterial relaxation when IDO1 uses $H_2O_2$ to convert L-Trp to cis-WOOH. A prerequisite for such a scenario is that GPx and Prx react slower with cis-WOOH than $H_2O_2$. Given the relatively high expression of GPx4 in resistance mesenteric arteries (Table 1), and its ability to reduce $H_2O_2$ and organic hydroperoxides with rate constants in the order of $10^5$–$10^7$ $M^{-1} s^{-1}$ [17,29], we examined the reaction of purified rat GPx4 with L-Trp-derived hydroperoxides. Apparent second-order rate constants for the reaction of GPx4 with cis- and trans-WOOH were $1.0 \times 10^6$ and $1.6 \times 10^6$ $M^{-1} s^{-1}$, respectively (Fig. 2a, top panel). The value obtained for cis-WOOH was ~100-fold lower than that for the reaction of GPx4 with phosphatidylcholine hydroperoxide (PC-OOH).

GSH also reacted with cis- and trans-WOOH in the absence of GPx4 (Fig. 2a, bottom panel), indicating that non-enzymatic reduction of cis-WOOH by GSH could conceivably contribute to the regulation of cis-WOOH-mediated relaxation of resistance arteries if arterial GSH were present in millimolar concentration. We therefore determined the GSH content in aorta and mesenteric arteries from naïve and LPS-treated mice by LC-MS/MS. Aorta and mesenteric arteries contained ~0.6 and

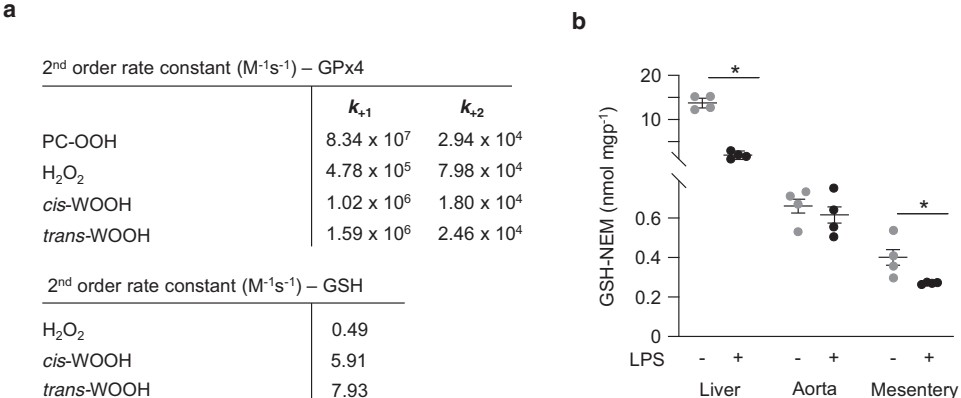

**Fig. 2 Reactions of *cis*- and *trans*-WOOH with GPx4. a** Upper panel shows the second-order rate constants for the reaction of *cis*- and *trans*-WOOH with GPx4. $k_{+1}$ is the rate constant for the reduction of the hydroperoxide, $k_{+2}$ is the cumulative rate constant for the regeneration of the reduced enzyme. The lower panel shows second-order rate constants for the non-enzymatic reaction of *cis*- and *trans*-WOOH with GSH. $H_2O_2$ and phosphatidylcholine hydroperoxide (PC-OOH) were used for comparison. **b** GSH content in abdominal aorta and mesenteric arteries from naïve mice and animals treated with lipopolysaccharide (LPS). Summary data are shown as mean ± SEM, $n = 4$ independent experiments. Statistical analysis was performed using two-tailed Mann–Whitney tests; *$p \leq 0.05$. Exact p-values are as follows: $p = 0.0286$ (liver, naïve vs LPS treated) and $p = 0.0286$ (mesentery, naive vs LPS treated). Source data are provided as a Source data file.

0.4 nmol mgp$^{-1}$ GSH, respectively, compared with ~15 nmol mgp$^{-1}$ GSH in liver (Fig. 2b). Treatment with LPS significantly decreased GSH concentrations in liver and mesenteric arteries (Fig. 2b). Assuming that endothelial and smooth muscle cells account for 74% of the mass of resistant arteries[30] and 1 g of tissue has a volume of 1 mL, the estimated average GSH concentration in resistance arteries of LPS-treated mice is ~540 µM. As GSH-dependent reactions of the reductive part of the cycle are rate-limiting ($k_2 \ll k_1$; see Fig. 1c), the low arterial concentrations of GSH can reasonably be expected to slow down the overall catalytic activity of GPx4. This observation and the comparatively slow reaction of GPx4 with *cis*-WOOH suggest that the GSH/GPx4 system does not play a major role in regulating arterial concentrations of *cis*-WOOH, consistent with the observed modest relaxation of resistance arteries from LPS-treated mice in response to BCNU.

**Reduction of *cis*-WOOH by Prx**. We next directed our attention to Prx4, the most abundant Prx in resistance arteries from LPS-treated mice (Table 1). *cis*- and *trans*-WOOH oxidized His-tagged Prx4 with an estimated rate of $k = \sim 10^3 M^{-1} s^{-1}$ at 5 °C as assessed by redox blotting (Supplementary Fig. 3a-b). This value is ~4 orders of magnitude lower than that for the reaction of Prx4 with $H_2O_2$[31] and argues against Prx4 playing a major role in the metabolism of *cis*-WOOH, especially as Prx4 is thought to be present predominantly in the endoplasmic reticulum and extracellular space[32], whereas IDO1 is a cytosolic enzyme and its product, *cis*-WOOH, likely requires a transporter to cross membranes. As Prx2 is predominantly located in the cytosol[32] where *cis*-WOOH is formed and its molecular target, PKG1α, is also present, Prx2 was selected for further detailed investigations.

We first considered assessing the reaction of Prx2 with ʟ-Trp-derived hydroperoxides using stopped-flow fluorescence spectrometry, a well-established method that follows the decrease in intrinsic Trp-derived fluorescence to monitor Prx2 redox state[33]. Unfortunately, however, the intrinsic fluorescence of WO(O)H[34] interferes with this assay. We therefore used redox blotting[35] to estimate the rate constant for the reaction of His-tagged Prx2 (5 µM) with *cis*-WOOH (5–40 µM) or *trans*-WOOH (2–16 µM). The time-dependent disappearance of Prx2 monomer was determined for *cis*-WOOH (Fig. 3a, b) and $k_{obs}$ calculated by single exponential equation (Fig. 3c). $k_{obs}$-values were linearly

dependent on the hydroperoxide concentration, and the second-order rate constant determined as $k = 3.8 \pm 0.5 \times 10^3 M^{-1} s^{-1}$ at 25 °C (Fig. 3c), in good agreement with previously published rate constants for the reaction of Prx2 with amino acid hydroperoxides[19]. Compared with *cis*-WOOH, *trans*-WOOH reacted faster with Prx2 so that experiments were performed at 5 rather than 25 °C (Fig. 3d, e). Linear fitting for $k_{obs}$ *versus* hydroperoxide concentration provided a second-order rate constant of $k = 1.1 \pm 0.3 \times 10^4 M^{-1} s^{-1}$ at 5 °C (Fig. 3f). Comparing the reactivity of Prx2 (5 µM) with *cis*-WOOH (32.5–130 µM) and *trans*-WOOH (6.5–26 µM) at 5 °C yielded $k$-values of $2.2 \times 10^3$ and $1.4 \times 10^4 M^{-1} s^{-1}$, respectively (Supplementary Fig 4a-e). Dimer-to-monomer ratios remained unchanged when: (i) NEM-pretreated His-tagged Prx2 was incubated with *cis*-WOOH or *trans*-WOOH; (ii) Prx2-C51S in which the peroxidatic cysteine residue is mutated to serine was used; and (iii) His-tagged Prx2 was incubated with *cis*-WOH or *trans*-WOH (Supplementary Fig. 5a-d).

To confirm the above data, we carried out competition experiments between Prx2 and thioredoxin reductase (TrxR, known to reduce organic hydroperoxides[36,37]) for reaction with *cis*- and *trans*-WOOH. Purified rat TrxR and increasing amounts of His-tagged Prx2 were used, and the initial rates of NADPH oxidation followed at 25 °C. As thioredoxin was not included, the reductive cycle of TrxR reducing Prx2 could not occur. TrxR reacted with *cis*- and *trans*-WOOH with rate constants of $2.1 \times 10^4$ and $1.8 \times 10^4 M^{-1} s^{-1}$, respectively (Supplementary Fig. 6a). NADPH oxidation was linear for approximately the initial 30 s of the reaction. Estimating the rate constants for the reaction of Prx2 with *cis*- and *trans*-WOOH from competition experiments (Supplementary Fig. 6b) yielded values of $\leq 2.1 \times 10^4$ and $1.8 \times 10^4 M^{-1} s^{-1}$, respectively, in agreement with the values obtained from redox blotting. These results were validated twofold. First, we confirmed by Coomassie staining that the Prx2 used was initially reduced (Supplementary Fig. 6c). Second, we confirmed that reaction with TrxR converted *cis*- and *trans*-WOOH (50 µM) to *cis*- and *trans*-WOH, respectively (Supplementary Fig. 6d). The alcohols were the only products detected and accounted for 62 and 84% of the loss of *cis*- and *trans*-WOOH, respectively. Together, these data indicate that Prx2 reacts with and reduces *cis*-WOOH and *trans*-WOOH with substantially lower efficacy compared with $H_2O_2$.

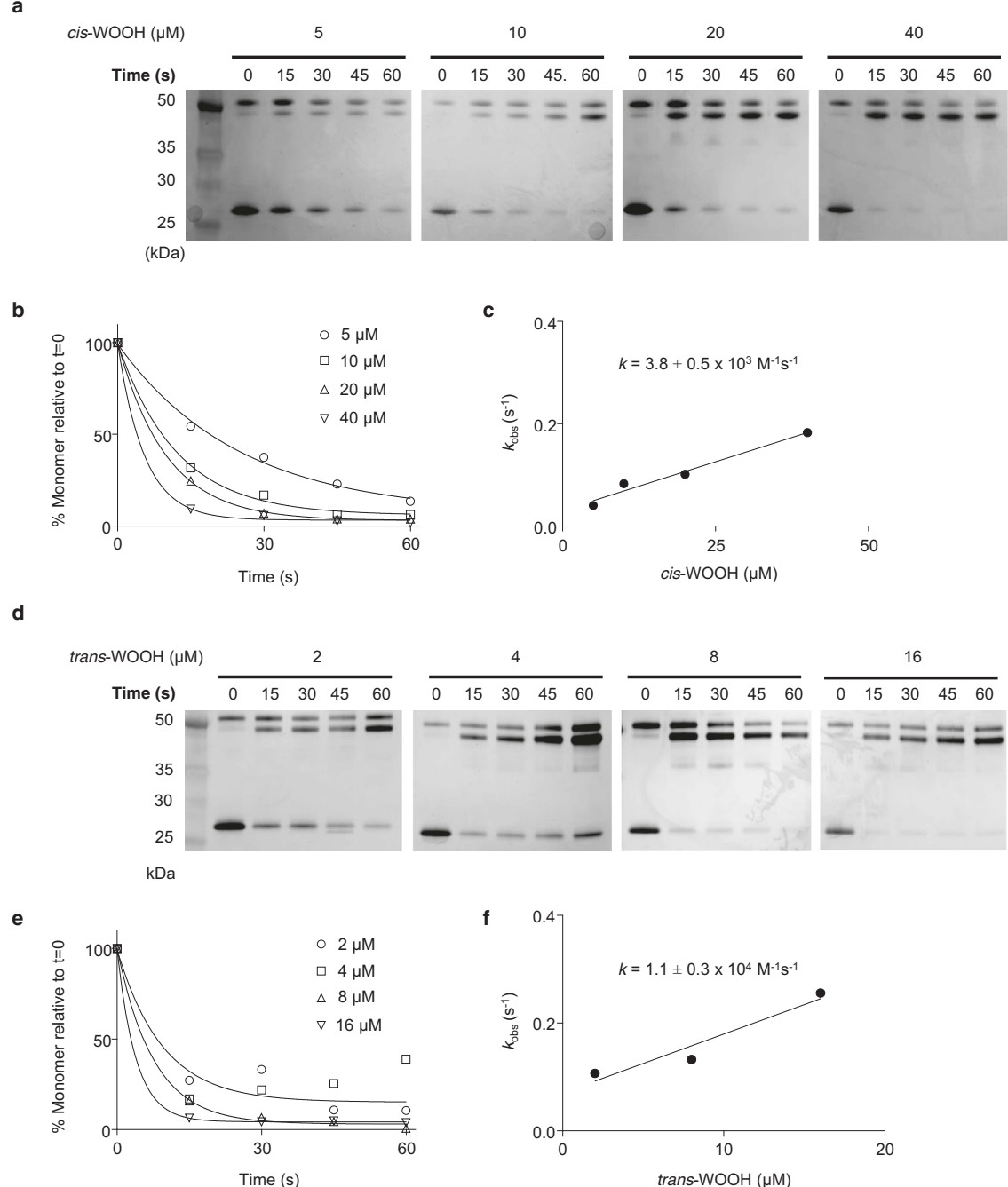

**Fig. 3 Estimation of the rate constant for the reaction of *cis*- and *trans*-WOOH with His-tagged wild-type Prx2. a** Reaction of Prx2 (5 µM) with *cis*-WOOH (5–40 µM) at 25 °C. After incubation, *N*-ethylmaleimide (NEM, 30 mM) was added, and the samples subjected to non-reducing 4–12% SDS-PAGE followed by silver staining of the gels. **b** Time course of monomer disappearance with increasing concentrations of *cis*-WOOH. **c** Determination of second-order rate constant for the reaction or Prx2 with *cis*-WOOH ($3.8 \pm 0.5 \times 10^3$ M$^{-1}$s$^{-1}$ at 25 °C). **d** Reaction of Prx2 (5 µM) with *trans*-WOOH (2–16 µM) at 5 °C. After incubation, NEM (30 mM) was added, and the samples subjected to non-reducing 4–12% SDS-PAGE followed by silver staining of the gels. Uncropped blots are provided in Source data. **e** Time course of Prx2 monomer disappearance with increasing concentrations of *trans*-WOOH. **f** Determination of the second-order rate constant for the reaction of Prx2 with *trans*-WOOH ($1.1 \pm 0.3 \times 10^4$ M$^{-1}$s$^{-1}$ at 5 °C). Rate constants values are mean ± SEM. Results shown are representative of three independent experiments. Uncropped blots and source data are provided as a Source data file.

**Stereospecificity in reactions of Prx2 with L-Trp-derived hydroperoxides**. The results from the redox blotting experiments suggested that Prx2 reacts preferentially with *trans*- than *cis*-WOOH. Supporting this idea, exposure of His-tagged Prx2 (20 µM) to increasing concentration of *cis*-WOOH at 5 °C for 10 s only modestly altered the enzyme's thiol redox status (Fig. 4a) with ~75% Prx2 remaining as monomer at equimolar concentrations of the

hydroperoxide (Fig. 4b). By comparison, *trans*-WOOH and H$_2$O$_2$ induced more rapid thiol oxidation such that the enzyme was present predominantly in the dimeric form at hydroperoxide concentrations exceeding 10 µM (Fig. 4a, b). Similar comparative thiol oxidation patterns were observed upon exposure of untagged Prx2 to *cis*-WOOH, *trans*-WOOH, and H$_2$O$_2$ although Prx2 oxidation was overall greater (Supplementary Fig. 7a-b). Consistent with these

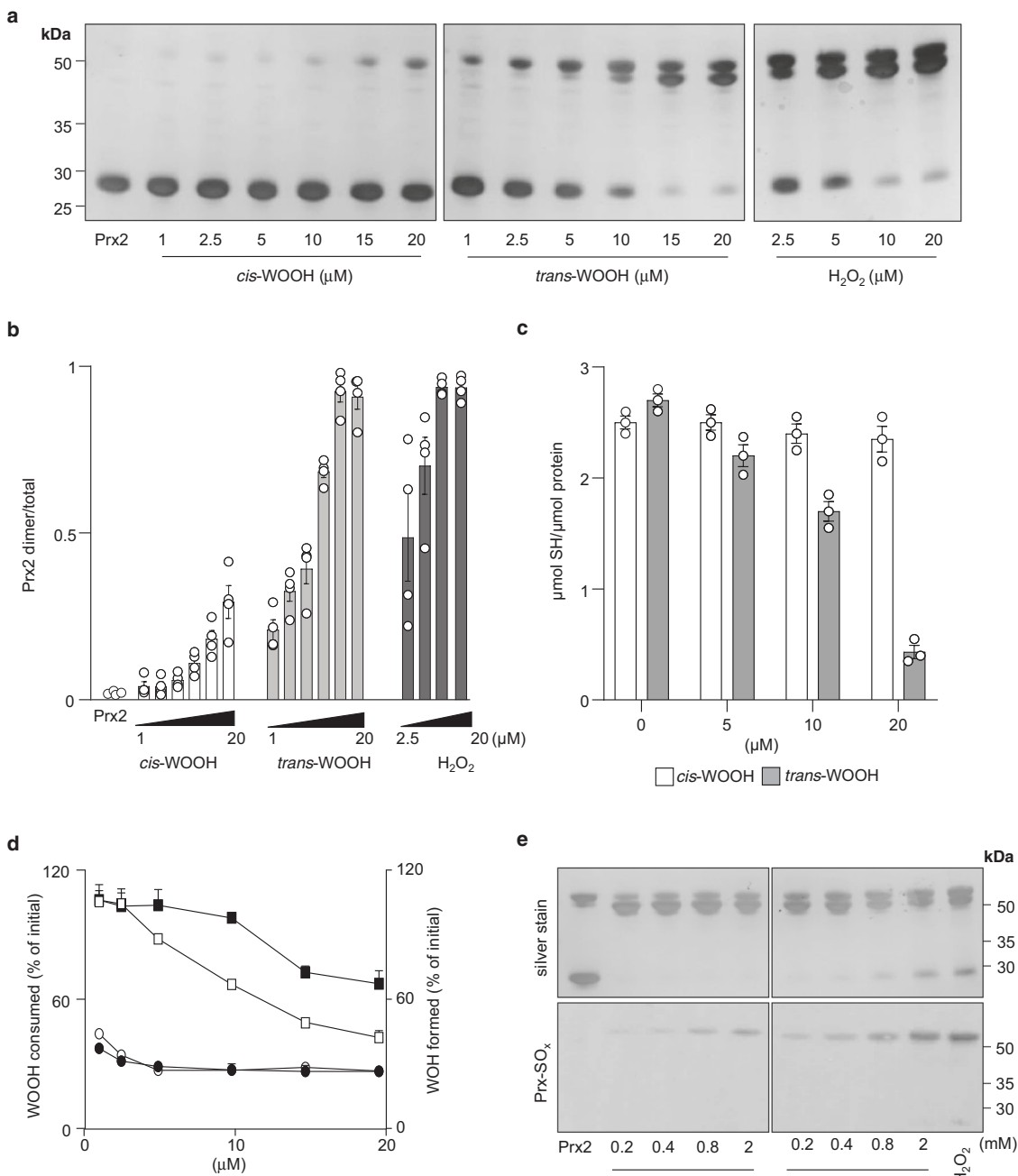

**Fig. 4 Differences in dimerization, thiol, and hydroperoxide consumption during the reaction of Prx2 with *cis*- versus *trans*-WOOH. a** Concentration-dependent reactions of *cis*- and *trans*-WOOH, and $H_2O_2$ (control) with pre-reduced His-tagged Prx2 (20 μM). Reactions were stopped after 10 s using NEM (50 mM). **b** Quantification of silver-stained Prx2 dimer and monomer bands in (**a**) with results expressed as ratio of Prx2 dimer-to-total Prx2, $n = 4$ independent experiments. **c** Quantification of free thiols after incubation of pre-reduced His-tagged Prx2 (20 μM) with *cis*-WOOH or *trans*-WOOH for 10 s before addition to 1 mM 5,5′-dithio-*bis*(2-nitrobenzoic acid), DTNB, $n = 3$ independent experiments. **d** Quantification of WOOH consumption and WOH formation in the presence of pre-reduced His-tagged Prx2 (20 μM) as determined by LC-MS/MS. Data are expressed as percentage of hydroperoxide consumed and corresponding alcohol formed relative to the initial hydroperoxide concentration, $n = 3$ independent experiments. **e** Absence of hyperoxidation of His-tagged Prx2 (20 μM) upon reaction with *cis*-WOOH or *trans*-WOOH. Samples were assessed in duplicates using non-reducing SDS-PAGE, with proteins on one gel silver stained and the other transferred to a polyvinylidene difluoride membrane before probing with an antibody specific for Prx-SO2/3. Data shown are representative of three (**a**) or two (**e**) separate experiments. Data in **b**, **c**, and **d** are shown as mean ± SEM. Uncropped blots and source data are provided as a Source data file.

observations, there was little change in the concentration of protein thiols when His-tagged Prx2 was exposed to *cis*-WOOH under the same conditions (Fig. 4c). This coincided with <50% reduction of *cis*-WOOH to *cis*-WOH despite the presence of a 20-fold molar excess of Prx2 (Fig. 4d). By contrast, exposure of Prx2 to *trans*-

WOOH resulted in the concurrent conversion of *trans*-WOOH to its respective alcohol (Fig. 4d). Also, *trans*-WOOH and $H_2O_2$, but not *cis*-WOOH, induced a concentration-dependent increase in Prx2 dimer containing one disulfide bond (Fig. 4a and Supplementary Fig. 7a, upper dimer band) followed by the accumulation of

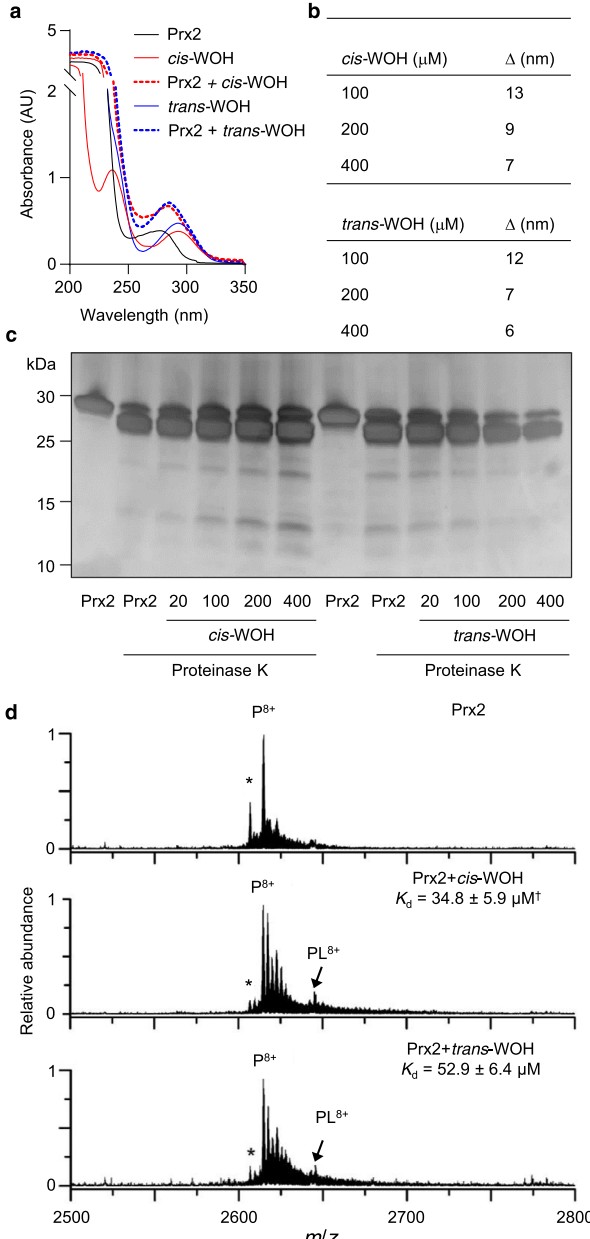

| cis-WOH (µM) | Δ (nm) |
|---|---|
| 100 | 13 |
| 200 | 9 |
| 400 | 7 |
| | |
| trans-WOH (µM) | Δ (nm) |
| 100 | 12 |
| 200 | 7 |
| 400 | 6 |

**Fig. 5 Differential binding and dissociation of *cis*- and *trans*-WOH to and from Prx2. a** Spectral shift analysis showed shifts from 277 nm (20 µM His-tagged Prx2 alone) to 280 (Prx2 + *cis*-WOH) and 288 (Prx2 + *trans*-WOH) indicative of isomer differences in protein binding. **b** Representation of data from (**a**) as delta of maximum absorption wavelength of both alcohols (100, 200 and 400 µM) in the presence of Prx2. **c** Silver staining following SDS-PAGE of samples reveals isomer-specific difference in degradation patterns of proteinase K digested pre-reduced His-tagged Prx2 (20 µM) after incubation with 20-400 µM *cis*- or *trans*-WOH. **d** Nanoelectrospray ionization mass spectrum of aqueous solutions containing pre-reduced His-tagged Prx2 and *cis*- or *trans*-WOH at 1:1 molar ratio reveals a shift of $\Delta m/z$ of 27.5 indicating non-covalent ligand-protein bonding. The dissociation constants ($K_d$) were obtained from the relative peak areas corresponding to unbound and ligand-bound protein ions reveal *cis*-WOH to have a higher affinity for Prx2 than *trans*-WOH. Traces/images are representative of 2 (**a**) or 3 (**c**, **d**) independent experiments. Data shown in (**b**) is mean of two independent experiments. $K_d$ values in (**d**) are given as mean ± SEM of three independent experiments. *Truncated form of Prx2 that is missing the first amino acid. †$p = 0.05$ compared with $K_d$-value for *trans*-WOH (one-tailed Mann–Whitney). Source data are provided as a Source data file.

Prx2 dimer with two disulfides (lower dimer band) whereas only the one disulfide form was seen with *cis*-WOOH. We also incubated His-tagged Prx2 (20 µM) with an excess of *cis*-WOOH or *trans*-WOOH for 5 min before SDS-PAGE/immunoblotting for hyperoxidized Prx2 monomer using a specific anti-Prx-SO$_{2/3}$ antibody. This revealed that *cis*-WOOH was less efficient in causing hyperoxidation of Prx2 than *trans*-WOOH (Fig. 4e), with the only signal detected in the disulfide dimer, suggesting the presence of a single interprotein disulfide bond with one peroxidatic cysteine residue hyperoxidized. These data indicate that Prx2 more efficiently reduces *trans*- than *cis*-WOOH.

**Stereospecific differences in the interaction of Prx2 with *cis*- and *trans*-WOOH.** To probe for potential differences in the interaction of Prx2 with *cis*- and *trans*-WOOH, spectral analysis of the complexes was performed using the more stable alcohol derivatives, *cis*- and *trans*-WOH, as surrogates for the hydroperoxides. Spectral analyses showed a red shift in peak absorbance from 277 to 280–286 nm upon addition of either alcohol to His-tagged Prx2 (Fig. 5a). The resulting absorption change was slightly but consistently different for *cis*- compared with *trans*-WOH, with the shift in absorption maximum being obvious at low ratios of alcohol to enzyme (Fig. 5b). This difference was not due to the His-tag, as exposure of untagged Prx2 to *cis*- or *trans*-WOH also caused different shifts in absorption (Supplementary Fig. 8a-b). Proteinase K-mediated proteolytic digest of Prx2 after incubation with *cis*- or *trans*-WOH revealed different degradation banding patterns for the two isomers (Fig. 5c). For example, intact Prx2 remaining after the digest increased with increasing *cis*-WOH concentrations whereas it decreased with increasing *trans*-WOH concentrations. Together, these results indicate that *cis*- and *trans*-WOH (and by implication *cis*- and *trans*-WOOH) bind to Prx2, albeit differently.

To obtain further evidence for the differential binding of *cis*-WOH and *trans*-WOH to Prx2, we attempted to determine the respective binding equilibrium through absorbance measurements. However, the comparatively strong absorption of Prx2 at 277 nm combined with its proximity to the absorption maxima of WOH and the low extinction coefficients of WOH, did not allow for such determination to be accurate. To overcome this, we instead employed 'native' mass spectrometry (MS) to directly and reliably determine ligand-protein binding constants[38]. In native MS, ligand-protein complex ions are formed directly from aqueous buffered solutions at near neutral pH, which can result in the retention of non-covalent solution-phase interactions in the gas-phase on the timescale of ion desolvation and detection (<1 s). Thus, native MS can be used to measure ligand-protein dissociation constants reliably and in excellent agreement with alternative solution-phase biochemical assays[39–42], particularly by using nanoscale ion emitter tips that reduce thermal destabilization of ligand-protein complexes during ion formation and transfer to the mass analyzer[39,43]. Importantly, native MS has shown to produce disassociation constants in excellent agreement with traditional spectroscopic methods[44].

A representative native mass spectrum of an aqueous solution containing His-tagged Prx2 (5 µM) is shown in Fig. 5d. An abundant ion was measured at $m/z$ 2614.8 that corresponds to the Prx2 monomer in the 8+ charge state (P$^{8+}$). The relatively low extent of ion charging for a protein of this size (21 kDa) is consistent with native mass spectra reported in the literature for proteins with similar mass. The addition of *cis*-WOH at a molar ratio of 1:1 to the solution resulted in the formation of a peak (PL$^{8+}$) that is shifted from the unbound Prx2 peak by $\Delta m/z$ of 27.5, which corresponds to a neutral mass of 220 Da (Fig. 5d middle); i.e., *cis*-WOH non-covalently binds to Prx2. The

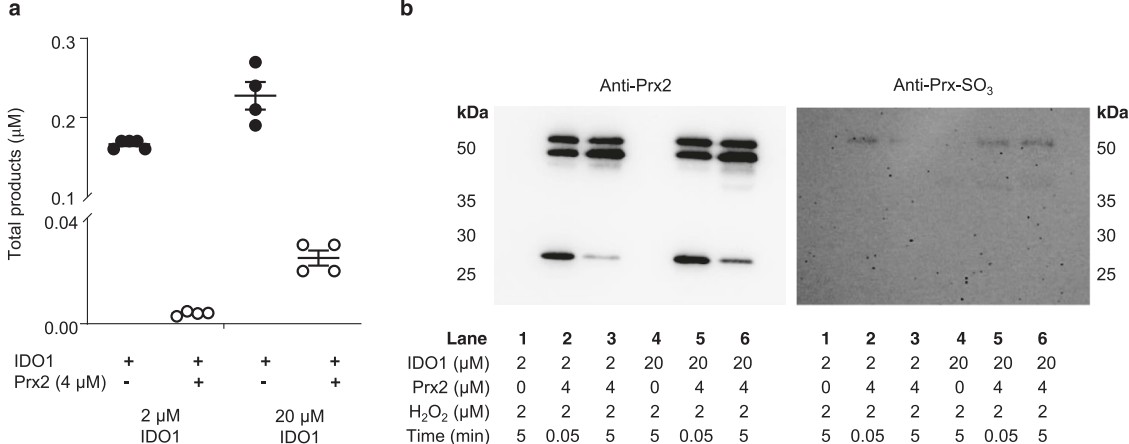

**Fig. 6 Utilization of $H_2O_2$ by IDO1 in the presence of Prx2. a** IDO activity assays (IDO1, 2 or 20 µM; $H_2O_2$ 2 µM; L-Trp; 100 µM) were carried out in the absence and presence of His-tagged Prx2 (4 µM). Reactions were incubated for 5 min at 25 °C and $H_2O_2$-derived Trp oxidation products (*cis*-WOOH and OIA) determined by LC-MS/MS. **b** Samples analyzed in (**a**) were collected and alkylated with NEM and probed by SDS-PAGE and western blotting for Prx2 oxidation state. Data are shown as mean ± SEM ($n = 4$ independent enzyme assays). Source data are provided as a Source data file.

addition of *trans*-WOH at the same molar ratio also resulted in a peak (PL$^{8+}$ in Fig. 5d bottom) that shifted by the same amount ($\Delta m/z$ of 27.5), indicating that *trans*-WOH also binds non-covalently to Prx2. The relative abundance of the ligand-bound to unbound Prx2 ion for *cis*-WOH was 14% higher than that for *trans*-WOH. By use of Eq. (2) (see "Methods"), the relative abundances of the ligand-bound to unbound ions, and the initial solution-phase concentrations of the protein and ligands, the dissociation constants ($K_d$) for *cis*-WOH (34.8 ± 5.9 µM) were significantly ($p \leq 0.05$) smaller than those for *trans*-WOH (52.9 ± 6.4 µM) as determined using three different ligand concentrations. These data indicate that *cis*-WOH has a higher affinity for Prx2 than *trans*-WOH, which we infer is representative of the affinity of *cis*-WOOH and *trans*-WOOH with Prx2.

**IDO1 utilizes $H_2O_2$ for enzymatic oxidation of L-Trp in the presence of reduced Prx2.** Our recent observation that IDO1-expressing 'inflamed' arteries form *cis*-WOOH[9] implies that IDO1 can compete with GPx and Prx for $H_2O_2$. Using time-resolved absorption changes, a bimolecular rate constant of $8 \times 10^3 \, M^{-1} \, s^{-1}$ was reported for $H_2O_2$-mediated conversion of IDO1 to its compound II type ferryl species ($Fe^{4+} = O^2$)[45]. To obtain separate information on how rapidly IDO1 reacts with $H_2O_2$ in presence of L-Trp, IDO1 activity assays were carried out in the absence or presence of reduced Prx2. We first added 2 µM $H_2O_2$ to 2 µM IDO1 ± 4 µM fully reduced His-tagged human Prx2 and determined the extent of conversion of L-Trp (100 µM) to *cis*-WOOH and oxyindolylalanine (OIA), i.e., the specific products of the oxidative dioxygenase and peroxidase activity of IDO1, respectively[9]. As expected, the presence of Prx2 significantly decreased, but did not prevent, formation of *cis*-WOOH/OIA as determined by LC-MS/MS (Fig. 6a). We then repeated the experiment using a 5-fold molar excess of IDO1 over Prx2. Under these conditions, the presence of Prx2 decreased formation of *cis*-WOOH/OIA by 88% (Fig. 6a). Using this data, the competitive approach, and a rate constant of $k = 2 \times 10^7 \, M^{-1} \, s^{-1}$ for the reaction of Prx2 with $H_2O_2$, we estimated the rate constant for the reaction of IDO with $H_2O_2$ in the presence of L-Trp as $\sim 3.2 \times 10^5 \, M^{-1} \, s^{-1}$, i.e., substantially greater than the $8 \times 10^3 \, M^{-1} \, s^{-1}$ reported previously[45].

We next probed for Prx2 monomer and dimer as well as hyperoxidized Prx2 using non-reducing SDS-PAGE and western blotting of NEM-treated samples post LC-MS/MS analyses (Fig. 6b). This confirmed the presence of Prx2 monomer and absence of hyperoxidized Prx2 monomer (lanes 3 and 6) at the

end of the reaction, i.e., after formation of *cis*-WOOH/OIA. Increasing the molar ratio of IDO1 to Prx2 led to greater amounts of Prx2 monomer remaining (compare lane 6 versus lane 3 in Fig. 6b).

Finally, we carried out in silico analyses using different rate constants for the reaction of IDO with $H_2O_2$ and applying them to the experimental data obtained when IDO1 was present at five-fold molar excess over Prx2. The simulations predicted *cis*-WOOH/OIA formation to occur in association with a decrease in Prx2 dimerization with a rate of reaction between IDO1 and $H_2O_2$ of $10^5 \, M^{-1} \, s^{-1}$ (Supplementary Fig. 9a) but not $10^3 \, M^{-1} \, s^{-1}$ (Supplementary Fig. 9b).

**cis-WOOH oxidizes PKG1α in vitro and in vivo.** The above data indicate that IDO1 can utilize $H_2O_2$ for enzymatic oxidation of L-Trp to *cis*-WOOH in the presence of reduced Prx2, and that significant amounts of *cis*-WOOH may escape reductive inactivation by arterial GPx and Prx and therefore be available to react with its target protein PKG1α. To provide supporting evidence for this scenario in vivo, mice were treated with LPS to induce the expression of IDO1, the enzyme required for the conversion of Trp to *cis*-WOOH. L-Trp was then administered to LPS-treated mice via intravenous injection to acutely elevate production of endogenous *cis*-WOOH[9]. After perfusion of the circulatory system with NEM to preserve the thiol redox state, the oxidation state of the two major arterial peroxidases, Prx2 and Prx4, as well as PKG1α were assessed in mesenteric arteries by immunoblotting (Fig. 7a). Acute production of *cis*-WOOH in vivo resulted in significant oxidation of C42 in PKG1α (Fig. 7b) with no observable change in the thiol redox status of Prx2 and Prx4 (Fig. 7c, d) or evidence of high molecular weight complex formation for either Prx isoform (Supplementary Fig. 10).

Finally, we compared the extent of PKG1α dimerization induced by *cis*-WOOH versus $H_2O_2$. As Cys42 in PKG1α is highly sensitive to autoxidation, the thiol reducing agent TCEP was added in molar excess over PKG1α, as described previously by others[46]. Under these conditions, *cis*-WOOH caused greater PKG1α dimerization than the same concentrations of $H_2O_2$ (Fig. 7e, f). While exact rate constants cannot be extrapolated from this data, it suggests that the reaction between *cis*-WOOH and PKG1α is kinetically more favorable than the reaction of PKG1α with $H_2O_2$.

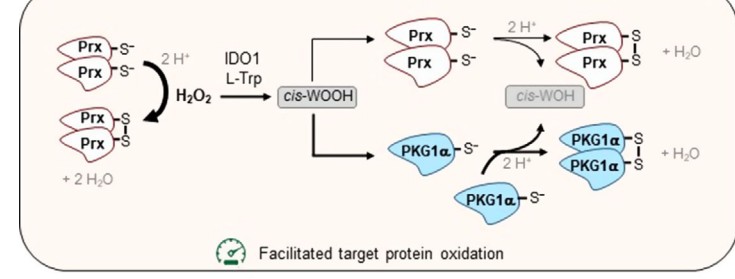

## Discussion

$H_2O_2$ has long been proposed to contribute to the relaxation of naïve resistance arteries[47,48]. Consistent with this, our results from arteries isolated from naïve mice indicate that this relaxation is regulated by arterial GPx and Prxs. What is new, however, is that the importance of such regulation diminishes significantly under inflammatory conditions where arterial relaxation is dependent on $H_2O_2$/IDO1-mediated conversion of L-Trp to $cis$-WOOH. We show further that the major peroxidase in mouse resistant arteries, i.e., Prx2, has a stereospecific preference to react with and reduce $trans$-WOOH compared with its $cis$-epimer and that the reaction with both Trp-derived hydroperoxides are substantially slower than the corresponding reactions with $H_2O_2$. In part, the comparatively slower reaction of Prx2 with $cis$-

**Fig. 7 Stereoselective oxidation of a target protein by *cis*-WOOH. a** Representative (*n*=6) immunoblot showing the effects of endogenous *cis*-WOOH formation on the redox states PKG1α, Prx2, and Prx4 in mesenteric resistance arteries isolated from LPS-treated mice after one-minute infusion of vehicle or L-Trp (8 mM) via the jugular vein. **a–e** Summary data of experiments performed in (**a**): L-Trp significantly increased dimerization of PKG1α ($p = 0.0173$) (**b**) without altering the thiol redox state of Prx2 (**c**) or Prx4 (**d**). Data in **b**, **c**, and **d**, are shown as mean ± SEM ($n = 6$ independent experiments) and statistical analysis performed using two-tailed Mann–Whitney tests with *$p \leq 0.05$. **e** Purified recombinant PKG1α (3.3 μM) was incubated with 200 μM *cis*-WOOH or $H_2O_2$ for 5 min in the presence of 200 μM tris(2-carboxyethyl)phosphine (TCEP) in 25 mM phosphate buffer pH 7.0 at 25 °C. At the specified time points, an aliquot was collected, and the reaction stopped by the addition of 125 mM maleimide. PKG1α redox state (proportion monomer and dimer to total) was assessed using non-reducing SDS-PAGE and silver staining. **f** Percentage of PKG1α monomer and dimer of total PKG1α was determined by densitometry for the samples analyzed in (**e**). **g** Model of stereospecific hindrance of reductive inactivation of *cis*-WOOH facilitating downstream target protein oxidation. Accordingly, $H_2O_2$ is required for but does not itself cause, oxidation of target signaling proteins. Rather, $H_2O_2$ is utilized by IDO1 in the presence of Prx to form an amino acid-derived hydroperoxide in a stereospecific manner, i.e., preferential formation of *cis*- over *trans*-WOOH. *cis*-WOOH and *trans*-WOOH differently bind to and react with Prx and downstream protein targets that may limit exposure of the hydroperoxide moiety of *cis*-WOOH to the catalytic cysteine residue in Prx. If so, this impedes Prx oxidation/*cis*-WOOH reduction, thereby facilitating *cis*-WOOH reaction with downstream target proteins such as PKG1α. Interaction with Prx2 leads to comparatively enhanced reduction of *trans*-WOOH, thereby impeding reaction of *trans*-WOOH with target proteins. We propose that this stereospecific evasion of reduction by *cis*-WOOH allows for greater spatial regulation of redox signaling such as inter-cellular signaling, and in cells with high Prx concentrations. See Discussion for further details. Uncropped blots and source data are provided as a Source data file.

WOOH than *trans*-WOOH can be explained by stereospecific differences in binding and dissociation of the hydroperoxides to Prx2. The observed stereospecific hindrance of reductive inactivation of *cis*-WOOH by peroxidases facilitates engagement of this epimer in the oxidation of its arterial target, PKG1α. Together, our results support a model for in vivo redox signaling by $H_2O_2$ whereby this primordial hydroperoxide facilitates the enzymic formation of an amino acid-derived hydroperoxide, *cis*-WOOH, that achieves oxidative activation of its key target PKG1α in part via stereospecific 'escape' of reductive inactivation by Prx and GPx.

Our evidence for the proposed new model is supported by multiple complementary experiments and techniques including in vivo experiments, in vitro work with isolated arteries and purified enzymes, and in silico kinetic modeling of experimental data. First, stimulation of arterial *cis*-WOOH formation in vivo by intravenous administration of L-Trp to LPS-pretreated mice at a time when endothelial IDO1 expression is maximal, caused dimerization of PKG1α in the mesenteric arteries in the absence of both detectable changes to the thiol redox state of Prx2 and Prx4, and formation of high molecular weight complexes involving Prx2 and Prx4. These results suggest that in the presence of fully intact cellular reducing systems and vasodilatory pathways, in vivo formation of *cis*-WOOH is associated with selective oxidation of PKG1α.

Second, pharmacological blockade of GPx or Prx elicited relaxation of naïve resistance arteries that was dependent on the presence of Cys42 in PKG1α, a known target for $H_2O_2$ and *cis*-WOOH. In IDO1-expressing mesenteric arteries from LPS-treated mice that produce *cis*-WOOH, relaxation after blockage of Prx or GPx was attenuated, while relaxation to exogenously added $H_2O_2$ remained unchanged. As IDO1-expressing 'inflamed' arteries generate more $H_2O_2$ than naïve arteries[9], these results suggest that the observed diminished importance of Prx/GPx in regulating relaxation was likely due to a switch in the 'signaling' oxidant from $H_2O_2$ to *cis*-WOOH rather than a diminished capacity of 'inflamed' arteries to metabolize $H_2O_2$. Our kinetic studies indicate that the rate constant for the reaction of IDO1 with $H_2O_2$ in the presence of L-Trp is $\sim 10^5 M^{-1} s^{-1}$. This is at least an order of magnitude faster than the rate constant reported previously for $H_2O_2$-mediated formation of the IDO1 compound II type ferryl species[45], although the approach used in that study did not measure formation of the initial ferric-(hydro) peroxo intermediate ($Fe^{3+}=O(H)OH$) and hence did not give direct information on the rate of reaction of IDO1 with $H_2O_2$. Reaction of heme peroxidases with $H_2O_2$ to form compound I is

usually faster than formation of compound II from compound I[49,50]. Importantly our data show that even in the presence of Prx2, IDO1 can react efficiently with $H_2O_2$ to produce *cis*-WOOH from L-Trp.

Third, our kinetic experiments and estimation of arterial concentrations of GSH suggest that in vivo, *cis*-WOOH is not likely to be reduced effectively by either GPx4, Prx2, or Prx4, i.e., peroxidases abundant in mouse resistance arteries. While the reaction of GPx4 with *cis*-WOOH in vitro is fast, i.e., second-order rate constant $\sim 10^6 M^{-1} s^{-1}$, its biological importance in resistance arteries appears limited under inflammatory conditions. This is because LPS decreased arterial GPx4 and the estimated low arterial GSH concentration is well below the millimolar concentrations required for GPx4 catalytic activity to be maintained[51–53]. Also, such low GSH concentrations favor GPx4 to be localized to membranes rather than the cytosol[54] where both IDO1 and *cis*-WOOH's target, PKG1α, are located. The observed low arterial GSH concentration further suggests that direct reduction of *cis*-WOOH by GSH is unlikely a major metabolic route for the elimination of the hydroperoxide in arteries, although our in vivo experiments (Fig. 1) do not rule out the possibility that GSH/GPx4 may play a minor role in the regulation of arterial concentrations of *cis*-WOOH.

Compared with GPx4, the reaction of Prx4 with *cis*-WOOH appeared negligible, making the participation of this peroxidase in the metabolism of *cis*-WOOH unlikely, especially as Prx4 is located predominantly in the extracellular space and the endoplasmic reticulum[55,56]. Of the most abundant cytosolic Prx[57], Prx1 was less abundant in resistant arteries than Prx2, suggesting that in vivo, Prx2 is the most relevant isoform in terms of *cis*-WOOH metabolism. Prx2 gave a second-order rate constant of $3.8 \times 10^3 M^{-1} s^{-1}$ for *cis*-WOOH and $1.1 \times 10^4 M^{-1} s^{-1}$ for *trans*-WOOH. This is close to two orders of magnitude slower than the reaction of GPx4 with *cis*- or *trans*-WOOH, and three to four orders of magnitude slower than the reaction of Prx2 with $H_2O_2$.

Separate lines of evidence indicate that there is a stereospecific difference with which Prx2 reacts with *cis*- and *trans*-WOOH. Examining Prx2 dimerization, loss of Prx2 thiols and the rate of *cis*-WOH formation separately and consistently supported the notion Prx2 reduced *cis*-WOOH less effectively than *trans*-WOOH, particularly at low molar ratios of substrate to enzyme which most likely reflects the situation in vivo. By comparison, $H_2O_2$ oxidized Prx2 more rapidly as assessed by all parameters. Further evidence for the stereospecific difference in reaction rate of the two epimers with Prx2 comes from the observation that at

the concentrations tested only *trans*-WOOH gave rise to hyper-oxidized Prx2. It is not clear whether this can be explained simply by the different extent of oxidation, or whether it supports a mechanism of intra-dimer cooperativity, whereby the oxidation by *trans*- but not *cis*-WOOH of the first peroxidatic thiol to sulfenic acid in the Prx2 homodimer enhances the rate of oxidation of the second peroxidatic site[58].

The slower reduction of *cis*- compared with *trans*-WOOH by Prx2 could be explained by different binding at the active site or binding at distinct sites. The latter scenario may effectively require the substrate to associate and dissociate multiple times before oxidation of the catalytic thiol occurs. The observed changes in absorption in the presence of Prx2 indicate that both epimers bind to the enzyme, a feature not previously reported for other substrates of Prx2. More importantly, native mass spectrometry showed that the *cis*-isomer has a higher affinity for Prx2 than the *trans*-isomer, indicating that once bound the *cis*-isomer does not dissociate as readily as the *trans*-isomer. Together, our data reveal a stereospecific interaction of *cis*-WOOH with Prx2 that helps explain its 'escape' of reductive inactivation. This is observed especially at low molar ratios of substrate to enzyme, a situation that likely reflects in vivo conditions where endogenous concentrations of *cis*-WOOH are low.

Previous studies have shown that the expression of 2-Cys Prx isoforms is lower in resistance compared with conduit arteries, and that this may enhance the likelihood of $H_2O_2$-mediated relaxation of resistance arteries[22]. Our work suggests that this difference is explained largely by differences in the expression of Prx4 rather than Prx2 between arterial beds. We also observed that LPS treatment of mice increased Prx4 in resistance arteries with no significant change in the expression of Prx2. The expression of Prx4 has been reported to increase in immune cells during experimental sepsis[59]. Interestingly, we observed LPS-induced endotoxemia to change the thiol redox state of arterial Prx2 and 4 only modestly, as assessed by redox immunoblotting. We used NEM to perfuse mice at euthanasia to preserve the in vivo thiol redox status of proteins, as validated by red blood cells' Prx2 being almost fully reduced. By comparison, arterial Prx2 and 4 were substantially oxidized, with about half of these enzymes present as oxidized disulfide dimers.

We were unable to calculate the exact rate of reaction of *cis*-WOOH with PKG1α as Cys42 in purified recombinant PK1α is highly sensitive to autoxidation in the absence of reductants. We therefore performed semi-quantitative kinetic studies following the rate of PKG1α dimerization in response to *cis*-WOOH or $H_2O_2$. This suggested the reaction of PKG1α with *cis*-WOOH to be kinetically favored over that with $H_2O_2$, supporting the role of *cis*-WOOH in oxidative modification of PKG1α. In vivo experiments confirmed preferential oxidation of PKG1α by *cis*-WOOH in a complex system with intact reducing systems, supporting the view that *cis*-WOOH is a biologically relevant oxidant controlling vascular tone. While our studies increase our understanding of how $H_2O_2$ signals in arteries under inflammatory conditions, they do not help explain how $H_2O_2$ signals in naïve arteries.

There has been a long debate as to how $H_2O_2$ can effectively engage in thiol redox signaling considering its low tissue concentrations, rapid metabolism by cellular reducing system, and limited reactivity with most protein thiols. To overcome these intrinsic problems underlying redox signaling by direct reaction of $H_2O_2$ with the target protein[7,8], the redox relay[5,6], floodgate[1–3], and transient inactivation models[4] have been proposed as alternative models. The present work adds to the understanding of oxidant sensing and signaling as it provides an example of how these previously described models are not needed for *cis*-WOOH-mediated redox signaling in arteries. We therefore propose a novel model of $H_2O_2$-mediated redox signaling (Fig. 7g). In this model, $H_2O_2$ is utilized by the heme-containing enzyme IDO1 to generate an amino acid-derived hydroperoxide, *cis*-WOOH, that in turn, escapes from reductive inactivation and activate its target protein in a stereospecific manner.

The proposed model of $H_2O_2$-mediated redox signaling in inflammation has several novelties and advantages. First, it replaces the site of $H_2O_2$ formation with the location of IDO1 as the spatial determinant for the formation of the signaling oxidant. As IDO1 is not expressed constitutively, a biological advantage of this redox signaling system is that it can be switched on in specific biological conditions (inflammation) and at a specific location (e.g., arterial endothelium). Production of $H_2O_2$ could similarly be upregulated in a site-localized manner. Second, $H_2O_2$-dependent formation of an alkyl hydroperoxide attenuates hydroperoxide reactivity with cellular peroxide scavenging systems. Third, predominant formation of *cis*- over *trans*-WOOH by IDO1 introduces stereospecificity in redox signaling at the level of the signaling oxidant and we propose that this model of stereospecific redox signaling has likely evolved. This is because the ancestor *Ido1* gene has properties more similar to present day *Ido2* than to *Ido1*[60], and IDO2 has a lower stereospecificity and activity compared with IDO1 in oxidizing L-Trp to *cis*-WOOH[9]. Finally, formation of a stereospecific redox signaling molecule by $H_2O_2$/IDO1/L-Trp allows for stereoselective interaction with and oxidation of target proteins. This may extend beyond arteries and PKG1α. Indeed, unpublished data from our laboratory using a redox proteomics screen suggest the existence of a *cis*-WOOH-specific redox proteome, raising the possibility of stereospecific signaling by *cis*-WOOH, and possibly also *trans*-WOOH, under different biological conditions.

## Methods

**Animals and tissue collection.** All experiments involving mouse tissues complied with the ethical regulations of, and were approved by, the Animal Ethics Committees of the Garvan Institute of Medical Research/St Vincent's Hospital and Sydney Local Health District. Arterial tissue was obtained from 8- to 12-week-old male C57BL/6J (Animal Resources Centre) and PKG1α C42S knock-in mice (C57BL/6N background)[9]. Animals were housed in rooms with a 12 h light/dark cycle corresponding to sunrise and sunset of approximately 6 am and 6 pm, respectively. The room temperature was maintained between 20 and 26 °C with humidity set between 40 and 70%. Where indicated, mice were subjected to endotoxemia by intraperitoneal injection of 7.5 mg kg$^{-1}$ body weight LPS (0111:B4, Sigma-Aldrich L4130) for 16 h. Mice were anesthetized using isoflurane and tissue collected after cardiac puncture, exsanguination, and perfusion using sterile PBS in the absence and presence of NEM as indicated. Once collected, arterial tissue was placed into ice-cold Krebs-Henseleit buffer (Sigma-Aldrich K3753, supplemented with 2.5 mM $CaCl_2$, 10 mM EDTA, 2.1 mg mL$^{-1}$ sodium bicarbonate and adjusted to pH 7.4) and dissected free of adipose and connective tissue. Stomachs were also collected and immediately snap frozen for further analysis.

**Myography.** Third-order mesenteric arteries (157.1 ± 3.6-μm inner diameter) were taken between the base of the mesenteric arterial arcade and the gut wall[61]. Immediately after dissection, arteries were mounted onto 25 μm wires (Multiwire myograph system, Model 610 M, Danish Myo Technology) submerged in Krebs-Henseleit buffer. To establish passive tension arteries were 'normalized' to 90% of the internal circumference obtained at 100 mm Hg[62]. Once passive tension was established, arteries were contracted with EC$_{80}$ noradrenaline and tested for endothelium-dependent relaxation (relaxation defined as the percentage reversal of noradrenaline-imposed tone) using acetylcholine (ACh, 10 μM). The endothelium was considered functional if ACh elicited relaxation responses of ≥70% (naïve)[63] or ≥40% (LPS-treated)[64]. Bath chambers containing arteries were then washed out with fresh Krebs-Henseleit buffer and arteries equilibrated for 30 min in Krebs-Henseleit buffer containing $N^{\omega}$-nitro-L-arginine methyl ester (L-NAME, 100 μM) and indomethacin (10 μM). Arteries were constricted with EC$_{80}$ noradrenaline and constriction allowed to stabilize for ~20 min. Where indicated arteries were constricted using 120 mM KCl. Once arterial tone was stable, arteries were exposed to 1,3-*bis*(2-chloroethyl)-1-nitrosourea (BCNU, 100 μM), auranofin (AUR, 300 nM), cumulative increases in $H_2O_2$ concentrations (1–300 μM) or the respective vehicle controls, EtOH, DMSO, and $H_2O$. Maximal responses were observed and recorded after ~3 min and results expressed as percentage relaxation. AUR and BCNU concentration-dependent relaxation response curves were carried out using the indicated concentrations and appropriate vehicle control.

**Estimation of Prx and GPx content in mouse arteries by immunoblotting.** Abdominal aortas and mesenteric beds were isolated from naïve C57BL/6J mice or mice treated with LPS as described above and then pooled separately (4 aortas and 6 mesenteric arterial beds for each pool), except for Prx1 expression that was assessed using arteries from three separate naïve and LPS-treated animals. The arteries within each pool were then pulverized in liquid $N_2$, suspended in radio-immunoprecipitation assay (RIPA) buffer with the addition of 2x EDTA-free cOmplete Protease Inhibitor Cocktail (Sigma-Aldrich) and homogenized on ice using a polytron homogenizer (Heidolph ST1) at 350 rpm, in 30 s intervals over a period of 5 min. DTT (5 mM) was added to the crude homogenate before samples were incubated at room temperature for 10 min, centrifuged at 17,000 × g for 5 min at 4 °C, and the protein concentration determined in the resulting supernatant using the BCA protein assay. Laemmli sample buffer (50 mM Tris-HCl, pH 6.8, 2% SDS, 0.1% bromophenol blue, 10% glycerol) containing DTT (100 mM) was added to arterial samples to dilute to the desired concentration. To assess the concentrations of endogenous Prx1-6, GPx1 and 4, and PKG1α in the supernatants of these arterial homogenates, recombinant proteins were used to generate respective standard curves. Briefly, commercial recombinant mouse Prx3-6 and GPx1 (Abbexa Ltd) were reconstituted in ultrapure water at 200 ng mL$^{-1}$. Recombinant human Prx1, Prx2, PKG1α, and rat GPx4, prepared as described below, were diluted to 200 ng mL$^{-1}$ in Laemmli sample buffer. Human Prx1, Prx2, and PKG1α share 95%, 93%, and 99% and rat GPx4 shares 92% of identity with mouse Prx1, Prx2, PKG1α, and GPx4, respectively (Protein BLAST). Increasing amounts of tagged recombinant Prx3-6 and GPx1, were added to a matrix of thoracic aorta homogenate to create a standard curve; tagged Prx1 and untagged recombinant Prx2, GPx4, and PKG1α were diluted into Laemmli sample buffer containing DTT alone. Protein standards were adjusted as described in Figure Legends to ensure that the respective standard curve covered the endogenous amounts of Prx, GPx, and PKG1α. Stomach homogenate (100 mg tissue per mL) prepared from a single C57BL/6J mouse was used to assess the presence of GPx2 in naïve arteries only as we were unable to source recombinant GPx2.

Prior to electrophoresis, arterial samples and recombinant protein standard curve samples were heated (60 °C, 10 min), centrifuged (17,000 × g, 10 min at room temperature), and resulting supernates subjected to SDS-PAGE using 4–20% Mini-PROTEAN TGX (Bio-Rad) using Tris/glycine/SDS running buffer and then transferred onto 0.2 µm nitrocellulose membranes (Bio-Rad) at 2.5 amp for 10 min using the Trans-Blot Turbo Blotting System (Bio-Rad). The membranes were blocked for 2 h in skim milk (10%, wt/vol) in Tris-buffered saline with the addition of TWEEN 20 (0.05%) (TBST). Blocked membranes were incubated overnight at 4 °C in skim milk (5%, wt/vol) and TBST with either Prx1 (1:1000 dilution, vol/vol, AdipoGen, YIF-LF-MA0214), Prx2 (1:2000 dilution, vol/vol, Sigma, R8656), Prx3 (1:500 dilution, vol/vol, Abfrontier, LF-PA0030), Prx4 (1:2000 dilution, vol/vol, Abcam, ab184167), Prx5 (1:2000 dilution, vol/vol, Abcam, ab180587), Prx6 (1:2000 dilution, vol/vol, Abcam, ab59543), GPx1 (1:2000 dilution, vol/vol, Abcam ab62204), GPx2 (1:500 dilution, vol/vol, Sigma-Aldrich SAB2700206, GPx4 (1:2000 dilution, vol/vol, Abcam ab125066), PKG1α (1:1000 dilution, vol/vol, ENZO Life Sciences, ADI-KAP-PK005) antibody or mouse polyclonal anti-actin (1:5000 dilution, vol/vol, MP Biomedicals, 8691001) antibody. The membranes were then probed for 1 h at room temperature with polyclonal goat anti-rabbit horseradish peroxidase-conjugated IgG (1:5000 dilution, vol/vol, Dako P0446) for Prx, GPx, and PKG1α, or polyclonal goat anti-mouse horseradish peroxidase-conjugated IgG (1:5000 dilution, vol/vol, Dako P0447) for actin. Protein bands were detected through enhanced chemiluminescence, as per manufacturer's instructions (ECL detection reagent and Hyperfilm ECL, GE Healthcare Biosciences) and analyzed by densitometry using Image Studio Lite (LI-COR, v5.2.5). The concentration of endogenous Prx, GPx, and PKG1α was estimated by comparison of band intensity with the standard curve and converted to "fmol/µg of tissue" using respective molecular weights of recombinant and endogenous proteins (obtained from UniProt). Arterial content of GPx2 in naïve arteries was assessed qualitatively by comparison of band intensities to that of increasing amounts of stomach homogenate.

**Estimation of the redox state of arterial Prxs and PKG1α.** Naïve and LPS-injected mice were anesthetized using isoflurane and blood drawn from the left ventricle into a syringe containing EDTA and maleimide (100 mM). The mouse's circulatory system was then perfused via the left ventricle with 12 mL phosphate buffer (PBS) containing NEM (100 mM) to alkylate thiols and remove residual blood. The abdomen was then opened, and abdominal aortas cleaned in situ of fat and connective tissue before being excised and snap frozen in 10 µL of maleimide (200 mM). Mesenteric tissue was collected into Krebs-Henseleit buffer containing NEM (100 mM) and mesenteric arterial beds (superior mesenteric artery to terminal third and fourth-order arterioles) then dissected and cleared of fat and connective tissue before snap freezing in 10 µL maleimide (200 mM). Individual abdominal aortas and mesenteric bed arteries were pulverized in liquid $N_2$ and homogenized in Laemmli sample buffer. Samples were subjected to non-reducing electrophoresis, immunoblotting for PKG1α, Prx2, and Prx4 and analyzed as described above. The thiol redox state was assessed by the intensity of dimer band to total (monomer plus dimer) band intensity.

To evaluate the effect of endogenous arterial formation of cis-WOOH on the redox state of Prx2, Prx4, and PKG1α, an incision was made in the neck of

anesthetized mice 16 h after administration of LPS, and the right jugular vein exposed and dissected from circumferential connective tissues. Mice were then subjected to intravenous (right jugular vein) infusion of HPLC-purified L-Trp (~8 mM, final blood concentration) or vehicle control. After 1 min, blood and tissues were collected as described above and immunoblotted to determine the thiol redox state of PKG1α, Prx2, and Prx4.

**Synthesis of tryptophan-derived hydroperoxides and hydroxides.** Tricyclic Trp-derived hydroperoxides (cis- and trans-WOOH) were prepared as described recently[9,34]. LC-MS/MS and HPLC were used to confirm the purity of the isolated hydroperoxides and alcohols. For LC-MS/MS, alcohols and hydroperoxides were diluted to 1 µM in LC-MS grade water (cis-WOH and cis-WOOH) or 30% DMSO (trans-WOH and trans-WOOH) prior to injection onto an Agilent 1290 UHPLC system connected to an Agilent 6490 triple-quadruple mass spectrometer with an ESI source operating in positive-ion mode. The alcohols and hydroperoxides were eluted using a 5 µm Luna C18(2) column (30 × 2.10 mm; Phenomenex) at a flow rate of 0.15 mL min$^{-1}$ using the following gradient: 100–95% mobile phase A (0.1% acetic acid in $H_2O$) from 0 to 6 min using mobile phase B (0.1% acetic acid in acetonitrile) and 5–100% mobile phase B from 6 to 8 min. At 8.6 min the gradient switches back to 100% mobile phase A for 1.4 min to re-equilibrate the column (total run time of 10 min). Mass spectrometer parameters were as follows: gas temperature = 250 °C; gas flow = 20 L min$^{-1}$; nebulizer pressure = 35 psi; sheath gas heater = 325 °C; sheath gas flow = 12 L min$^{-1}$; capillary voltage = 3500 V; fragmentor voltage = 380 V and cell accelerator voltage = 5 V. Hydroxides and hydroperoxides were checked for contamination by other Trp-derived metabolites by multiple reaction monitoring (MRM). Settings for the target analytes were (parent ion → fragment ion); Trp ($m/z$ 205 → 146, RT = 6.3 min) with collision energy (CE) = 17 V; kynurenine ($m/z$ 209 → 146, RT = 3.9 min) with CE = 10 V; NFK ($m/z$ 237 → 192, RT = 5.0 min) with CE = 5 V; cis-WOOH ($m/z$ 237 → 203, RT = 4.0 min) with CE = 5 V; trans-WOOH ($m/z$ 237 → 203, RT = 3.0 min) with CE = 5 V; cis-WOH ($m/z$ 221 → 175, RT = 2.5 min) with CE = 15 V; trans-WOH ($m/z$ 221 → 175, RT = 1.9 min) with CE = 15 V and oxyindolylalanine ($m/z$ 221 → 175, RT = 4.8 and 5.2 min) with CE = 15 V. For HPLC, alcohols and hydroperoxides were diluted to ~10 µM in LC-MS grade water (cis-WOH and cis-WOOH) or 30% DMSO (trans-WOH and trans-WOOH) prior to injection onto a 5 µm Luna C18 100 Å column (15 × 4.6 mm; Phenomenex) at a flow rate of 0.7 mL min$^{-1}$ using 0.1% acetic acid in $H_2O$ as mobile phase. Confirmation of purity was done by the detection of a single peak corresponding at the indicated retention time (RT) to either trans-WOOH (RT 11.9 min), cis-WOOH (RT 16.8 min), trans-WOH (RT 7.7 min), and cis-WOH (RT 11.2 min). The purities obtained for cis-WOOH and trans-WOOH were >95% by LC-MS/MS and >99.5% for cis-WOH and trans-WOH (LC-MS/MS and HPLC).

**Quantification of $H_2O_2$, cis-WOOH, trans-WOOH, cis-WOH, and trans-WOH.** Solutions of $H_2O_2$ were prepared from stock immediately before use and their concentrations determined spectrophotometrically by reaction with horseradish peroxidase (Sigma-Aldrich) to form compound I, using $\Delta\varepsilon_{403nm} = 54,000$ M$^{-1}$ cm$^{-1}$[65]. Solutions of cis-WOOH and trans-WOOH were prepared by dissolving the powder in ultrapure water and DMSO (30% in ultrapure water), respectively. Their concentrations were determined by monitoring the reaction with potassium iodide (6%) and EDTA (1 mM) in deoxygenated methanol:acetic acid (2:1, v/v) at 25 °C to produce triiodide and using $\varepsilon_{358nm} = 29,700$ M$^{-1}$ cm$^{-1}$[66]. For some experiments, concentration of cis-WOOH and trans-WOOH were determined using spectrophotometry and their extinction coefficients, $\varepsilon_{295nm} = 2750$ and 2460 M$^{-1}$ cm$^{-1}$, respectively, as described in detail[34]. Solutions of cis-WOOH and trans-WOH were prepared by dissolving the powder in ultrapure water or and DMSO (30% in ultrapure water), respectively. For quantification by LC-MS/MS, compounds were separated on a 5 µm Luna C18(2) column (30 × 2.10 mm; Phenomenex) eluted at 0.15 mL min$^{-1}$ as outlined earlier. Analytes were quantified against corresponding standards. Data acquisition was performed using the Agilent Mass Hunter Data Acquisition software v.B.06.00 build 6.0.6025.0 and analysis was performed using Mass Hunter Qualitative Analysis v.B.06.00 build 6.0.633.10 service pack 1.

**Determination of rate constant for the reaction of GPx4 with cis-WOOH and trans-WOOH.** GPx4 was isolated from rat testis mitochondria as described previously[67] and stored in 25 mM Tris-HCl, 0.5 M KCl, 10% (v/v) glycerol, 5 mM 2-mercaptoethanol, pH 7.5. For enzymatic and non-enzymatic reactions 0.1 M KH$_2$PO$_4$/K$_2$HPO$_4$ pH 7.8 containing 50 µM DTPA, 0.1% (v/v) Triton X-100, 160 µM NADPH and 0.6 IU mL$^{-1}$ glutathione reductase and reduced glutathione (GSH) at 2, 3, and 4 mM was used as buffer. Reactions were carried out at room temperature and initiated by the addition of the peroxidic substrate (20 µM). cis-WOOH and trans-WOOH were dissolved in 0.1 M KH$_2$PO$_4$/K$_2$HPO$_4$ pH 7.8 and 50 µM DTPA/DMSO (50/50, vol/vol), respectively. Phosphatidylcholine hydroperoxide (PC-OOH, dissolved in methanol) was used as positive control. The GPx4 concentration used was 66.8 nM (specific activity: 130 µmol min$^{-1}$ mgp$^{-1}$ with 2.5 mM GSH) for cis- and trans-WOOH, and 33.4 nM for PC-OOH. The decrease in 340 nm absorbance was determined from progression curves of NADPH oxidation in a Cary 50 Scan spectrophotometer and $\varepsilon = 6220$ M$^{-1}$ cm$^{-1}$ was used for calculations. Kinetic analyses were carried out from single progression curves of

NADPH oxidation at different GSH concentrations[11,16]. Change in absorbance versus time was used to calculate substrate concentration and rate constant at time intervals of 5 s. The rate of the reaction catalyzed by GPx4 was corrected for the rate of the reaction with the same substrate, at the same concentrations, but with the enzyme buffer. No correction is needed only for PC-OOH.

Apparent rate constants for GPx4 were calculated by the simplified Dalziel equation:

$$E/v_0 = \phi_0 + \phi_1/[ROOH] + \phi_2/[GSH]$$

where $\phi_0$ is the reciprocal of the turnover number (0 for GPx4 as for the other GPx) and $\phi_1$ and $\phi_2$ are the Dalziel coefficients, equivalent to the reciprocal second-order rate constant of the peroxidatic and reductive steps of the reaction, respectively. Absorbance data at 340 nm from the progression curves of NADPH oxidation, obtained in the presence of GSH and triggered by $2 \times 10^{-5}$ M peroxidic substrate, were used to calculate pseudo first-order rate constants. Peroxidic substrate concentrations at each interval time (1 s) were calculated using an extinction coefficient of 6220 $M^{-1}$ $cm^{-1}$. The slope ($k'$) of the straight line obtained by plotting $\ln[A]$ vs time (s) is $k[B]_0$ and represents the pseudo first-order rate constant, where $[A]$ is the concentration of the peroxidic substrate and $[B]_0$ is the concentration of GSH kept constant by GSSG reductase activity. Second-order rate constants for reaction of the peroxidic substrates with GSH were calculated by plotting pseudo first-order rate constants against the GSH concentration. The same results were obtained by fitting the second-order equation.

**Quantification of glutathione in mouse arteries.** Arterial concentrations of GSH and oxidized glutathione (GSSG) were determined principally as described previously[68,69]. Isotopically labeled GSSG and GSH-NEM, generated from commercial glycine-labeled GSH ([glycine 1,2-$^{13}C_2$, $^{15}N$]-GSH), was a generous gift from Prof A. Kettle, University of Otago, Christchurch, New Zealand. Liver, abdominal aortas, and mesenteric beds were isolated from NEM-perfused naïve or LPS-treated C57BL/6J mice as described above. The tissues (~2 and 20 mg for arteries and liver, respectively) were then pulverized in liquid $N_2$, suspended in 40 parts of cold PBS containing NEM (20 mM), and homogenized on ice using a polytron homogenizer (Heidolph ST1) at 350 rpm, in $5 \times 30$ s intervals over a period of 5 min. An aliquot of the homogenate was taken, centrifuged at $17,000 \times g$ for 5 min at 4 °C, and the protein concentration determined in the resulting supernatant using the BCA assay. Livers and aortas were diluted 20- and 2-fold in PBS containing NEM (20 mM) before addition of internal standards (labeled GSH-NEM, 0.1 μM; labeled GSSG, 0.2 μM) and trichloroacetic acid (30%). After centrifugation at $5000 \times g$ for 5 min at 4 °C, the upper layer was collected and stored at $-80$ °C. For analysis, the samples were diluted 5-fold in ultrapure water and then injected into an Agilent 1290 UHPLC system connected to an Agilent 6490 triple-quadrupole mass spectrometer with an ESI source operating in positive-ion. GSH-NEM and GSSG were separated on an Agilent Poroshell 120 column (2.7 μm; $2.1 \times 100$ mm) at 40 °C. The mobile phase consisted of formic acid (0.1%) in water (mobile phase A), and formic acid (0.1%) in acetonitrile:isopropanol (1:1, v/v). The separation was carried out at a linear gradient at flow rate of 0.2 mL $min^{-1}$ from 100% mobile phase A to 30% mobile phase B (0–15 min), followed by wash with 100% mobile phase B (15–20 min) then returned to the initial conditions for equilibration (20–30 min). Mass spectrometer parameters were as follows: gas temperature = 170 °C; gas flow = 19 L $min^{-1}$; nebulizer pressure = 20 psi; sheath gas heater = 400 °C; sheath gas flow = 12 L $min^{-1}$; capillary voltage = 3500 V. Quantification of GSH-NEM and GSSG was conducted by multiple reaction monitoring with fragmentor voltage and cell accelerator voltage set at 380 and 4 V, respectively, and using extracted ion chromatogram of GSH-NEM ($m/z$ 433 → 304, RT 5.5 min) and GSSG ($m/z$ 613 → 484, RT 3.5 min). For the isotopically labeled internal standards, the settings were GSH-NEM ($m/z$ 436 → 307) and GSSG ($m/z$ 619 → 490). Under these conditions, the lower limits of detection for GSH-NEM and GSSG were 0.0125 and 0.005 pmol, respectively. The calibration curves were then generated for each analyte, ranging from 0.0125 to 5 pmol for GSH-NEM and 0.005–1.25 pmol for GSSG, using labeled GSH-NEM (0.05 pmol) or GSSG (0.1 pmol), respectively. The area ratio of the analyte peak and the internal standard was plotted against the amount of analyte, and the concentration of GSH-NEM and GSSG in the tissues determined by interpolating the area ratio of the sample and internal standard against the corresponding calibration curve. For mouse arteries, the amount of GSSG was below the detection limit. Data were acquired using the Agilent Mass Hunter Data Acquisition software v.B.06.00 build 6.0.6025.0 and processed using Mass Hunter Qualitative Analysis v.B.06.00 build 6.0.633.10 service pack 1.

**Expression and purification of wild type and mutant human Prx.** His-tagged wild type human Prx2, C51S human Prx2, and wild type human Prx1 and Prx4 were cloned into expression vector pET-17b (Prx1), pET-28a (wild type and C51S Prx2), or pETDuet-1 (Prx4), before transformation of *E. coli* BL21 (DE3) Competent Cells (New England Biolabs) and purification of the proteins from positive clones as described previously[70,71], with minor modifications. Briefly, cell pellets resuspended in lysis buffer (50 mM Tris-HCl pH 7.4, 100 mM NaCl, 10% glycerol, 0.1 mM phenylmethylsulfonyl fluoride, 1x cOmplete protease inhibitor cocktail, 0.1 mg $mL^{-1}$ lysozyme, 10 mM imidazole, 25 μg $mL^{-1}$ DNAse I) were incubated for 10 min on ice before sonication (cycles of 20 s "on" and 40 s "off") on ice for

2 min using a digital sonicator (Branson Digital Sonifier) set at 30% amplitude. The extracts were clarified by centrifugation at $26,640 \times g$ for 45 min at 4 °C, and the resulting supernate from one pellet incubated under slow rotation at 4 °C with 1.5 mL Ni-NTA agarose beads (Qiagen) pre-equilibrated with lysis buffer. After 2 h, the flow through was collected by gravity, and the resin washed with 5 vol washing buffer (50 mM Tris-HCl pH 7.4, 100 mM NaCl, 20 mM imidazole), followed by protein collection with 6 vol elution buffer (50 mM Tris-HCl pH 7.4, 100 mM NaCl, 250 mM imidazole). The collected fractions were diluted in Laemmli sample buffer containing DTT, subjected to 4–12% SDS-PAGE (NuPAGE Bis-Tris gel, Invitrogen), followed by silver staining of the gel according to the manufacturer's recommendation (Pierce$^{TM}$ Silver Stain kit). Fractions with highest protein purity were combined and dialyzed extensively against 50 mM Tris-HCl buffer pH 7.4 containing 100 mM NaCl and 100 μM diethylenetriaminepentaacetic acid (DTPA) using SnakeSkin dialysis tubing (Thermo Scientific, 10 MWCO). Glycerol (10%) was then added and protein, at concentrations <100 μM, stored at 4 °C. Where indicated, His-tags were cleaved by incubating Prx2 (1–2 mg $mL^{-1}$) for 15 h at 4 °C in the presence of Factor Xa (1:400, New England Biolabs) and CaCl$_2$ (1 mM), and the tags removed by Ni-NTA agarose beads. Protein concentration was determined by the BCA assay method as per manufacturer's instructions using bovine serum albumin as standard.

**Expression and purification of human PKG1α.** Expression and purification of human PKG1α was performed according to published methods[72,73] with the following modifications. FreeStyle$^{TM}$ 293-F cells (Gibco® R79007) were grown in serum-free medium (Invitrogen Gibco FreeStyle$^{TM}$ 293 Expression Medium, 12338026) in a CO$_2$- and humidity-controlled shaking incubator (Kuhner shaker, 140 rpm) for at least 5 passages prior to transfection. Cells were diluted to $3 \times 10^6$ cells $min^{-1}$ using fresh pre-warmed medium before pcDNA 3.1(+)PKG1α −TEV-His-V5 (6 μg $mL^{-1}$) was added. Stable cationic polymer polyethyleneimine (9 μg $mL^{-1}$) was added 5 min later as transfection reagent and cells cultured for 24 h before dilution (1:1, v/v) with pre-warmed medium containing sodium valproate (4.4 mM). Cells were harvested after further 72 h of culture by centrifugation ($1000 \times g$) for 5 min at 37 °C, resuspended in ice-cold lysis buffer (25 mM sodium phosphate buffer containing 300 mM NaCl, 2x EDTA-free protease inhibitor, 5 mM 2-mercaptoethanol, 10% glycerol, pH 7.0) and frozen in liquid $N_2$. Cells were gently lysed by four cycles of freeze (liquid $N_2$) thawing (tap water), and the lysate clarified by centrifugation at $24,000 \times g$ for 10 min at 4 °C. The resulting supernatant (~40 mg protein) was incubated at 4 °C with 1 mL Ni-NTA resin for 3 h. The flow-through fraction was collected by gravity, and the resin washed with 10 column volumes washing buffer (25 mM sodium phosphate buffer containing 1 M NaCl and 20 mM imidazole, pH 7.0). PKG1α was then eluted with elution buffer (25 mM sodium phosphate buffer containing 300 mM NaCl and 100–800 mM imidazole, pH 7.0). Aliquots of each fraction were subjected to reducing 4–12% SDS-PAGE (NuPAGE Bis-Tris gel, Invitrogen) and gels stained with InstantBlue$^{TM}$ Coomassie stain (Expedeon, ISB01L). Fractions containing PKG1α were pooled and dialyzed three times for 60 min each in 25 mM sodium phosphate buffer pH 7.0 containing NaCl (50 mM). PKG1α was further enriched by manual anion exchange using a HiTrap Q HP column (GE Healthcare) coupled to a syringe. After washing with 10 column vol 25 mM sodium phosphate buffer pH 7.0 containing 50 mM NaCl, proteins were eluted with 2 column vol of 25 mM sodium phosphate buffer, pH 7.0 containing increasing concentration of NaCl (100–700 mM). The fractions collected were subjected to reducing 4–12% SDS-PAGE in combination with InstantBlue$^{TM}$ Coomassie staining or immunoblotted for PKG1α as described above. PKG1α-containing fractions were again pooled and dialyzed three times for 60 min each against 25 mM sodium phosphate buffer pH 7.0 containing NaCl (100 mM) and 10% glycerol. The concentration of PKG1α was determined by BCA assay. Finally, purified PKG1α was treated with H6-TEV protease (New England Biolabs) to cleave the His-tag followed by removal of both free tag and H6-TEV protease using Ni-NTA beads.

**Dimerization of PKG1α by *cis*-WOOH and H$_2$O$_2$.** Purified recombinant PKG1α (3.3 μM) was incubated with *cis*-WOOH or H$_2$O$_2$ (200 μM) for 5 min at 25 °C in 25 mM phosphate buffer pH 7.0 containing tris(2-carboxyethyl)phosphine (TCEP, 200 μM). At indicated time points, aliquots were collected, and the reaction stopped by the addition of maleimide (125 mM). Samples were then subjected to 4–12% SDS-PAGE (NuPAGE Bis-Tris gel, Invitrogen) under non-reducing conditions using MOPS running buffer, followed by silver staining of the gels. Protein bands (PKG1α monomer and dimer) were quantified by densitometry using Image Studio$^{TM}$ Lite (LI-COR Biosciences v5.2.5).

**Pre-reduction of Prx2 and Prx4, and quantification of protein thiols.** Immediately prior to experiments, proteins were reduced with dithiothreitol (DTT) (5-fold molar excess of total thiol) for 1.5 h at 37 °C under an atmosphere of argon and with constant shaking at 350 rpm. Excess DTT was removed using two successive PD-10 SpinTrap columns (GE Healthcare) prewashed once with ultrapure water and then twice with ultrapure water pre-treated with bovine catalase (10 μg $mL^{-1}$) followed by equilibration with 5 washes of 50 mM phosphate buffer pH 7.4 pre-treated with bovine catalase (10 μg $mL^{-1}$) and containing 100 μM DTPA. In pre-treated water/phosphate buffer, bovine catalase was eliminated by

filtration through an Amicon Ultra-15 10 K filter (Millipore). The protein thiol content was determined spectrophotometrically by incubating 5 µL samples with 145 µL 5,5′-dithio-*bis*(2-nitrobenzoic acid) (DTNB, 1 mM in 50 mM Tris buffer pH 8.5 containing 1% SDS) in a 96-well plate for 10 min at room temperature. Samples were measured at 412 nm using a microplate reader (VersaMax, Molecular Devices) and the thiol content determined using $\varepsilon_{412nm} = 14{,}100 \, M^{-1} \, cm^{-1}$. Only pre-reduced proteins with 2–3 thiols per protein (Prx2), 1–2 thiols per protein (C51S Prx2), and 3–4 thiols per protein (Prx4) were used for subsequent experiments. For oxidation experiments, His-tagged wild type Prx2 (20 µM) was incubated with *cis*-WOOH or *trans*-WOOH (5, 10, and 20 µM) for 10 s at 5 °C in 50 mM sodium phosphate buffer pH 7.4 containing DTPA (100 µM) and free thiols then quantified immediately by DTNB as described above.

**IDO activity assays.** Recombinant human IDO1 was purified as previously described[9] and human recombinant His-tagged Prx2 was purified as outlined above. Reactions were set up in which IDO1 (2 or 20 µM) were incubated with $H_2O_2$ (2 µM) and HPLC-purified L-Trp (100 µM) in the presence and absence of pre-reduced His-tagged Prx2 (4 µM) in phosphate buffer (pH 7.4). Reactions were incubated for 5 min at 25 °C and 5 µL of reaction mixture was injected onto the LC-MS/MS. $H_2O_2$-derived Trp oxidation products (*cis*-WOOH and OIA) were determined and quantified as outlined above.

**Kinetic simulations.** Kinetic modeling was conducted using GEPASI 3.30 software (http://www.gepasi.org)[74]. Formation of $H_2O_2$-derived Trp oxidation products was simulated using the following conditions: Prx2, 4 µM; $H_2O_2$, 2 µM; L-Trp, 100 µM and 20 µM IDO1. The rate of $H_2O_2$ reaction with Prx2 was set at $k = 2 \times 10^7 \, M^{-1} \, s^{-1}$ and the rate of reaction between IDO1 and $H_2O_2$ was set at either $3.2 \times 10^5 \, M^{-1} \, s^{-1}$ or $8.0 \times 10^3 \, M^{-1} \, s^{-1}$.

**Dimerization of Prx by *cis*-WOOH and *trans*-WOOH.** Unless stated otherwise, His-tagged Prx2, Prx4, and untagged Prx2 (20 µM) were incubated with *cis*-WOOH, *trans*-WOOH (1–20 µM) or $H_2O_2$ (2.5–20 µM) for 10 s at 5 °C in 50 mM sodium phosphate buffer pH 7.4 pre-treated with catalase and containing DTPA (100 µM). Where indicated, *cis*-WOH or *trans*-WOH (1 mM) was added to Prx2 for 10 s, prior to exposure to *cis*-WOOH (1–20 µM) for 10 s. *N*-Ethylmaleimide (NEM, 50 mM) was then added immediately to alkylate remaining thiols, followed by Laemmli sample buffer. Samples were then subjected to 4–12% SDS-PAGE (NuPAGE Bis-Tris gel, Invitrogen) under non-reducing conditions using MOPS running buffer, followed by silver staining of the gels, and quantification of protein bands by densitometry using ImageJ[75]. For dimerization studies using higher concentrations of *cis*- and *trans*-WOOH, pre-reduced untagged Prx2 (5 µM) was incubated with *cis*-WOOH (32.5–130 µM) or *trans*-WOOH (6.5–26 µM) for 40 s at 25 °C (for *cis*-WOOH) or 4 °C (*trans*-WOOH) in 100 mM phosphate buffer pH 7.4 pre-treated with catalase and containing DTPA (100 µM). Aliquots were removed at 10 s intervals and NEM (20 mM) added immediately to alkylate remaining thiols. Zero time-point samples were produced by adding Prx2 to phosphate buffer containing NEM (20 mM). Samples were then diluted in phosphate buffer and subjected to SDS-PAGE and western blotting of Prx2 as outlined above. Protein bands (Prx2 monomer and dimer) were quantified by densitometry as described above.

For redox blotting kinetic assays, the extent of disappearance of Prx2 monomer in the presence of increasing concentrations of *cis*- or *trans*-WOOH was expressed before calculating $k_{obs}$ values. The latter were then plotted against hydroperoxide concentrations and the corresponding second-order rate constants determined from the slope of the corresponding linear fittings. For estimation of Prx4 rate constants, the percentage of Prx4 dimer ratio formed by equimolar *cis*-WOOH or *trans*-WOOH after 10 s was used to calculate an estimated rate constant using the following rate equation ($v = k \times [Prx4] \times [WOOH]$).

**Competition reaction of *cis*-WOOH with thioredoxin reductase (TrxR) versus Prx2.** NADPH (200 µM) and rat TrxR (0.2 µM) were equilibrated for 3 min in 50 mM HEPES buffer at 25 °C, before increasing amounts of *cis*-WOOH or *trans*-WOOH was added, and the reaction mixture incubated for a further 7 min. For experiments using His-tagged Prx2, increasing concentrations of pre-reduced His-tagged Prx2 were incubated with NADPH (200 µM) and TrxR (0.2 µM) for 3 min in 50 mM HEPES buffer at 25 °C to equilibrate, and then *cis*-WOOH or *trans*-WOOH (10 µM) were added, and the reaction incubated for a further 7 min. NADPH consumption was followed by determining the time-dependent decrease in 340 nm absorption using a spectrophotometer (Shimadzu UV-2550PC) at 25 °C. NADPH consumption was followed at 340 nm, and the rates of reaction TrxR with *cis*- or *trans*-WOOH determined over the initial 30 s and using the initial rate approach.

**Analysis of Prx2 hyperoxidation by *cis*-WOOH and *trans*-WOOH.** His-tagged Prx2 (5 µM) was incubated with *cis*-WOOH, *trans*-WOOH (0.2–2 mM) or $H_2O_2$ (0.2 mM) for 5 min at 25 °C in 50 mM sodium phosphate buffer pH 7.4, pre-treated with bovine catalase (10 µg mL⁻¹) and containing DTPA (100 µM). NEM (30 mM) was then added immediately, followed by Laemmli sample buffer. Samples were then subjected to 4–12% SDS-PAGE (NuPAGE Bis-Tris gel, Invitrogen) under

non-reducing conditions using MOPS running buffer, and the gel subjected to immunoblotting as described above by incubating the membranes overnight with anti-Prx-SO₂/₃ (1:1000 dilution, vol/vol, Abcam, Ab16830) at 4 °C. In some cases, a separate gel was subjected to silver staining.

**Prx2-mediated conversion of *cis*-WOOH and *trans*-WOOH to their respective alcohols.** His-tagged Prx2 (20 µM) was incubated with *cis*-WOOH or *trans*-WOOH (5, 10, and 20 µM) for 10 s at 5 °C in 50 mM sodium phosphate buffer pH 7.4 containing DTPA (100 µM). The reaction mixture was diluted 5-fold in ultrapure $H_2O$ and subjected to an Agilent 1290 UHPLC system connected to an Agilent 6490 triple-quadrupole mass spectrometer with an ESI source operating in positive-ion mode. Detection and quantification of *cis*-WOOH, *trans*-WOOH and their respective alcohols was carried out as outlined earlier. There was a delay of ~90 s between dilution and injection of samples onto the LC-MS.

**Binding of *cis*-WOH and *trans*-WOH to Prx2 assessed by UV-Vis spectroscopy.** His-tagged Prx2 or untagged Prx2 (20 µM) was incubated with *cis*-WOH or *trans*-WOH (100, 200 and 400 µM) for 10 s at 25 °C in 50 mM sodium phosphate buffer pH 7.4, before the solution was transferred to a quartz cuvette and spectral scans from 200 to 350 nm were recorded at room temperature using a Cary 100 UV-VIS spectrophotometer (Agilent Technologies).

**Effect of *cis*-WOH or *trans*-WOH on susceptibility of Prx2 to proteolysis.** His-tagged Prx2 (20 µM) was incubated at 25 °C in 50 mM sodium phosphate buffer pH 7.4 with *cis*-WOH or *trans*-WOH (20-400 µM) for 10 s before treatment with proteinase K (Astral Scientific, 1:10,000, wt:wt) for 30 min at room temperature. Loading buffer containing DTT (100 mM) was then added, and samples were subjected to 4–20% SDS-PAGE (Mini-PROTEAN TGX, Bio-Rad) using MOPS running buffer followed by silver staining of the gels.

**Native mass spectrometry.** Nanoelectrospray ionization emitters were fabricated with ~250 nm inner tip diameters from borosilicate glass capillaries (Harvard Apparatus, 1.2-mm o.d., 0.68-mm i.d.) using a microcapillary puller (Model P-97, Sutter Instruments, USA). Nanoelectrospray ionization emitters were coated with a mixture of gold and palladium simultaneously in batches of 20 emitters at a time (Scancoat Six, Edwards, UK). All mass spectrometry experiments were performed using a hybrid linear trap quadrupole and orbitrap mass spectrometer (LTQ Orbitrap XL; Thermo Fisher Scientific, USA). A voltage of +0.8 to 1.5 kV was applied to the nanoelectrospray ionization emitters relative to the heated capillary entrance to the mass spectrometer (270 °C) to initiate and maintain electrospray ionization. A maximum ion injection time of 500 ms was used throughout. Mass spectra were acquired for 2–3 min in triplicate using three different nanoelectrospray ionization emitters. For each mass spectrum, peak areas corresponding to the unbound (P) and ligand-bound protein (PL) were automatically integrated by an in-house software program entitled PL binding, which was implemented in MATLAB (2017a, The MathWorks, USA)[39]. An analytically derived equation that was used to obtain $K_d$ values for ligand binding to Prx2 was from the literature[39]:

$$K_{d,i} = \frac{\sum_n P^{n+}}{\sum_n PL_i^{n+}} \left( [L_i]_0 - \frac{\sum_n PL_i^{n+}}{\sum_n P^{n+} + \sum_i \sum_n PL_i^{n+}} [P]_0 \right) \quad (2)$$

where $K_{d,i}$ corresponds to the dissociation constant of the $i$th ligand ($L_i$); $P^{n+}$ and $PL_i^{n+}$ correspond, respectively, to the ion abundances of the unbound protein and $L_i$-bound complex; and $[L_i]_0$ and $[P]_0$ correspond to the initial concentrations of the ligands and protein, respectively.

**Statistical analysis.** No statistical methods were used to predetermine sample size. Data are expressed as mean ± SEM and analyzed for normality using the Shapiro–Wilk test. Statistical significance was assessed using the two-way ANOVA with Tukey's post hoc test for quantification of protein expression, or a one or two-tailed Mann–Whitney test (as indicated) for differences between control and intervention groups used in GSH content, LC-MS/MS, and native mass spectrometry experiments. For arterial relaxation experiments a one-way ANOVA with Holm–Sidak multiple comparison test or a two-tailed Mann–Whitney test was used as indicated in-text. Differences were considered significant at $p \leq 0.05$. The number of independent experiments performed is indicated by the individual data points shown.

**Reporting summary.** Further information on research design is available in the Nature Research Reporting Summary linked to this article.

## Data availability

All data generated in this study are provided in the Source data files. Information pertaining to recombinant and endogenous proteins was obtained from Uniprot. Accession numbers are as follows: Prx1 (P35700), Prx2 (Q61171), Prx2 (P20108), Prx4 (O08807), Prx5 (P99029), Prx6 (O08709), GPx1 (P11352), and GPx4 (O70325). Source data are provided with this paper.

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

## Acknowledgements

This work was supported by National Health & Medical Research Council Program Grant and Senior Principal Research Fellowship to R.S.; and an Early to Mid-Career Research Grant from NSW Health to C.P.S. W.A.D. thanks the Australian Research Council for financial support. We thank A. Peskin (University of Otago) for help with initial experiments on Prx2 expression, A.J. Kettle (University of Otago) for providing isotopically labeled GSSG and GSH-NEM, N.J. Bulleid (University of Glasgow) for providing Prx4 plasmid, and J. Chiu and P. Hogg (both University of Sydney) for providing purified TrxR. We also thank F.C. Meotti and J.C. Toledo Jr. (both University of Sao Paulo) for their generous advice on the kinetic modeling and D. Newington for his help with animal experiments.

## Author contributions

A.A. and R.S. jointly designed and supervised this work. R.F.Q. and S.M.Y.K. produced recombinant Prx. R.F.Q., N.M. and C.C.W. designed and/or carried out kinetic experiments with isolated Prx enzymes including the binding and decomposition of hydroperoxides and kinetic modeling. C.P.S. designed and carried out arterial relaxation studies. C.P.S. and K.W. designed and carried out studies to evaluate the thiol oxidation status of arterial proteins. C.P.S., K.W., R.R., R.F.Q., S.M.Y.K. and N.M. designed and carried out evaluation of arterial expression of peroxidases. R.F.Q. and R.R. designed and carried out GSH quantification. G.T.H.N. and W.A.D. designed and carried out the native mass spectrometry. A.R. and F.U. designed and performed GPx4 kinetic studies. S.G. and P.E. produced recombinant PKG1α and R.F.Q. and S.M.Y.K. carried out or analyzed kinetic experiments with recombinant PKG1α. A.A., S.M.Y.K., C.C.W. and R.S. designed and carried out IDO activity assays and TrxR kinetic experiments. All authors interpreted data. R.F.Q., K.W., C.P.S. and R.S. drafted the manuscript, and all authors critically reviewed the draft manuscript. A.A. and R.S. revised the manuscript with help from C.P.S. and all authors commented on and approved the submitted versions of the manuscript.

## Competing interests

The authors declare no competing interests.
