## [Peer Review File · Nature Communications]

Hydrogen peroxide signaling via its transformation to a stereospecific alkyl hydroperoxide that escapes reductive inactivationREVIEWER COMMENTS

Reviewer #1 (Remarks to the Author):

In this manuscript, the authors suggest that part of the vasodilator response after LPS treatment is mediated by an IDO-1 dependent formation of cis-WOOH, which subsequently oxidizes PKG1 α . Subsequent aspects address the stereospecificity of WOOH regarding its reaction of Prx. The manuscript essentially is a development and refinement of some previous work by the group establishing IDO-1 and WOOH as vasodilator principles in inflammation.

I have been asked by the editors to specific review the physiological part. I understand that much on these aspects has already been covered (in part) in previous publications by the group. Nevertheless, in the setting of the present work, some aspects require further consideration. In particular, I have the following specific question:

1) Throughout the vascular function studies, experiments were performed in the presence of L-NAME and indomethacin. Thus in a situation in which all vasodilators (with the exception of EDHF) are blocked. Moreover, L-NAME interferes with NOS-dependent superoxide production. This raises the obvious question how much of the reported effects is physiologically important, thus, in a condition where WOOH acts in concert with other vasodilators? Please therefore show the dilator responses to AUR, BCNU and H₂O₂ with and without LPS treatment in the absence of NOS/Cox-inhibition. In order to avoid additional animal experiments, ex vivo LPS treatment could be used.

2) LPS induces increases in some of the monomers (like Prx2 in RBCs). This might easily be a consequence of overoxidation. Is this so? This is an important aspect as dimer formation is obviously not the only sign of oxidation.

3) Given that WOOH and H₂O₂ both act through PKG oxidation, it is hard to understand that LPS, which is used to induce IDO-1 and to therefore to produce WOOH, does not impact on the H₂O₂-induced relaxation. In my opinion, this contradicts the authors' hypothesis.

4) On page 7 line 151 the authors aim to estimate the GSH concentration in arteries, but the calculation appears very shaky, given that the cytosolic GSH content is what matters. Vessels have a large extracellular space and a high connective tissue content. Given that the value is surprisingly low and that some further argumentation depends on this value, the authors should seek a way to accurately measure GSH cytosolic content (or better said the cytosolic volume).

5) On page 15, line 330/331 the authors suggest that the relaxation to H₂O₂ is controlled by GPx and Prxs, but this information is missing in the result section. Do BCNU and AUR indeed block the H₂O₂-induced relaxation?

6) What is the effect of auranofin on vascular tone in the PKG1 α C42S mice? This aspect is missing in the figures.

Reviewer #2 (Remarks to the Author):

In their manuscript "Stereospecific impediment of reductive inactivation of tryptophan-derived hydroperoxide: a new model for mammalian redox signaling by hydrogen peroxide" Queiroz et al present a novel and promising H₂O₂-dependent mechanism of redox signaling. The proposed model goes like this: H₂O₂ is enzymatically transformed into a tryptophan-derived hydroperoxide (WOOH), which in turn oxidizes the final target: protein kinase G1 α . The intermediate hydroperoxide would then "protect" the oxidant from the main peroxidases Prx and GPx. An idea akin to the redox relay model but with a hydroperoxide instead of sulfenic acid/disulfide intermediate.

The authors study in detail the reaction of human Prx2 and GPx4 with two epimers of WOOH, use an ex-vivo model of mouse artery relaxation combined with inhibition of the Trx and GSH systems to identify the involvement of Prx and GPx in interfering the dimerization of PKG1 α .

Although the idea is very interesting, there are a few important questions that need to be addressed before the manuscript is published:

1) Spectroscopic properties of WOOHs. A few characteristics of the reduction products (WOH) are mentioned, such as extinction coefficient and their UV spectra but I could not find any data on the

WOOHs. The question is important because a big portion of the kinetic study is made through the intrinsic fluorescence of Prx2. Questions that need to be answered are:

- a. Are cis and trans WOOH fluorescent?
- b. Do they interfere with the detection of Prx2 fluorescent changes, through FRET, quenching or inner filter effect?
- c. The same questions for the products cis- and trans-WOH.

On the same topic of spectroscopic properties, the results on Fig 4a and 4b are hard to understand, the WOH used as ligands have their own spectra, what is shown in 4a is in any way different from the sum of the spectra of protein + WOH? What is meant by the spectral shift? I am sure the authors know that the chromophore of Prx2 is a combination of its 2 Trp and its 7 Tyr residues, then, what is shifting? Can the authors show a differential spectrum? Can the binding equilibrium be followed through absorbance measurements instead of rather complex MS experiment as presented?

2) Prx1. The authors could not find a suitable antibody for assessing the presence of Prx1 and then they did not study its potential involvement in the redox signaling. This represents a major weakness in the manuscript, since Prx1 is the main cytosolic peroxiredoxin in most human cell types (Selvaggio, G. et al *Redox Biol.* 15, 297–315 (2018)).

3) The kinetic competing systems. According to the model, to oxidize PKG1 α , H₂O₂ is first transformed into WOOH by IDO1, therefore, there are two important kinetic systems in competition.

- a. Firstly, IDO1 competes with the peroxidases (Prx and GPx) for the reduction of H₂O₂. No mention of the reaction of the kinetics of IDO1 as a peroxidase is made in the manuscript nor I could find easily a rate constant that indicates the likelihood of H₂O₂ ending as a substrate of IDO1 instead of Prx and GPx, to support their claim the authors should show that the reaction of IDO1 is at least competitive.
- b. Secondly, the peroxidases Prx and GPx compete with PKG1 α for the reduction of WOOH. The authors dedicate a large fraction of the manuscript to describe the kinetics of cis and trans WOOH with Prx2 and GPx4 which they identify as the major peroxidases, but no parallel study is made on PKG1 α . Then, although WOOH appears to be a rather limited substrate, particularly for Prx2, it is not clear how fast is the reaction with the cysteine of PKG1 α .

4) Some serious problems in the kinetics as presented.

- a. The pseudo first order rate constants of GSH with H₂O₂ (and presumably the WOOHs) are overestimated, by dividing by [GSH] they yield second order rate constants above 100 M⁻¹ s⁻¹. One of the authors published a seminal paper reporting that constant as 0.87 M⁻¹ s⁻¹ by a much robust method (Winterbourn, C. C. & Metodiewa, D. *Free Radic. Biol. Med.* 27, 322–328 (1999)).
- b. Page 8, Line 176, and this is a red flag: "...and the dissociation constant (k₋₁) as -3.1 ± 0.8 s⁻¹". There are NO negative rate constants (or dissociation constants) and dissociation constants have units of concentration, not frequency.
- c. Figure 2d and S2b, the broken line profile of the kobs vs [WOOH] plot deserves an explanation, it is unexpected and hard to reconcile with a simple reaction scheme as proposed. What exactly happens at that 85 micromolar WOOH? Why the change in reaction order from zero to one?
- d. The K_m for GSH with GPx is "indefinite" according one of the authors (Ursini, F., et al *BBA - Gen. Subj.* 839, 62–70 (1985)), so how can it be GSH 4 fold lower than the K_m?
- e. It is not clear what functions the authors used to fit their experimental time courses, it says in p. 8 | 17 "the initial rapid decay fitted a single exponential equation" but clearly the initial decay cannot be separated from the slower increase so a sum of at least two exponential terms is needed. Also in p. 35 "The observed rate constants (kobs) for fluorescence decrease (initial phase) and increase (second phase) were determined by fitting the data to single exponential equations..." Does this mean the authors fitted the functions separating two segments of the time courses? If that is the case it may be part of the explanation for the unexpected behavior of Fig 2d and S2b.

One last question that is just terminology, it says in line 51 "H₂O₂ itself is a weak oxidant", not at all, oxidant strength is usually measured by the standard reduction potential and H₂O₂ is one of the strongest non-radical oxidants in biology, it is slow in many reactions, but not a weak oxidant.

Reviewer #3 (Remarks to the Author):

H₂O₂ was reported before to relax bovine arteries through PKG activation by both soluble guanylate cyclase (sGC)/cGMP-dependent and cGMP-independent thiol oxidation/subunit dimerization mechanisms. The idea is not novel. H₂O₂ was known to act as one of endothelium-derived hyperpolarizing factor in small mesenteric arteries in response to acetylcholine. However, the type of vascular cells affected by both inhibitors BCNU and AUR is unclear.

Other major comments include

1. The information for functional assay of arteries on myograph is insufficient. The concentration-response effects in response to either BCNU or AUR in noradrenaline-pre-constricted vessels are required although the relaxant effect of AUR is rather transient (biphasic). Were the arteries used without intact endothelium? Do both drugs also relax pre-constricted mouse aortas?
2. The additive effect of combined application of BCNU and AUR is required assuming both inhibitors are capable of increasing H₂O₂ generation in the arteries by blocking recycling of oxidized GPx and Prx enzymes, respectively in the arteries.
3. The effect of catalase should be tested on the relaxation induced by BCNU and AUR.
4. H₂O₂ should be used as the positive control in arteries from WT and PKG1alpha KO mice.
5. Does NO donor also produce less relaxation in arteries from PKG1alpha KO mice?
6. LPS may induce iNOS expression to reduce vascular tone while Fig. 1 f did not show such effect.
7. H₂O₂ may open vascular smooth muscle potassium channel to cause relaxation. The effect of BCNU and AUR should be tested in arteries constricted by elevated extracellular potassium ions for indication of the role of H₂O₂.
8. Vascular production of H₂O₂ should be measured after inhibitor treatment.
9. It should be tested whether inhibitor treatment can enhance acetylcholine-induced endothelium-dependent relaxation in small mesenteric arteries and such effect should be reduced by H₂O₂-degrading enzyme such as catalase in the myograph chamber.

POINT-BY-POINT RESPONSES

Reviewer #1 (Remarks to the Author):

In this manuscript, the authors suggest that part of the vasodilator response after LPS treatment is mediated by an IDO-1 dependent formation of *cis*-WOOH, which subsequently oxidizes PKG1 α . Subsequent aspects address the stereospecificity of WOOH regarding its reaction of Prx. The manuscript essentially is a development and refinement of some previous work by the group establishing IDO-1 and WOOH as vasodilator principles in inflammation.

I have been asked by the editors to specific review the physiological part. I understand that much on these aspects has already been covered (in part) in previous publications by the group. Nevertheless, in the setting of the present work, some aspects require further consideration. In particular, I have the following specific question:

1) Throughout the vascular function studies, experiments were performed in the presence of L-NAME and indomethacin. Thus in a situation in which all vasodilators (with the exception of EDHF) are blocked. Moreover, L-NAME interferes with NOS-dependent superoxide production. This raises the obvious question how much of the reported effects is physiologically important, thus, in a condition where WOOH acts in concert with other vasodilators? Please therefore show the dilator responses to AUR, BCNU and H2O2 with and without LPS treatment in the absence of NOS/Cox-inhibition. In order to avoid additional animal experiments, ex vivo LPS treatment could be used.

Response: We agree with the reviewer that the physiological relevance of results obtained with isolated arteries needs to be considered carefully. Notwithstanding this, however, key discoveries on arterial physiology and blood pressure regulation have been made by carrying out experiments where the nitric oxide and cyclooxygenase pathways have been blocked (*Nature* 1998;396:269; *ibid* 1999;401:493; *Br J Pharmacol* 2005;144:108; *Circ Res* 2011;108:566). We tested the physiological relevance of this pathway in our earlier work (*Nature* 2019;566:548, **Figure 1**). In addition, our original submission included data from *in vivo* experiments with Trp infusion into mice \pm LPS treatment (**Figure 2**). Under these conditions, none of the vasodilator pathways are blocked pharmacologically.

Figure 1. Regulation of arterial relaxation and blood pressure by IDO1-mediated Trp metabolism is dependent on cysteine residue 42 of PKG1 α . Blood pressure responses elicited by Trp and sodium nitroprusside (SNP) in LPS-treated wildtype (black, n=8 for Trp, n=4 for SNP) or PKG1 α C42S knock-in mice (grey, n=9 for Trp and SNP). Data are representative or mean \pm SEM. * $p \leq 0.05$ as determined by Kruskal Wallis with Dunn's test. Reproduced from *Nature* 2019;566:548.

Figure 2. a) Representative (n=6) immunoblot showing the effects of endogenous *cis*-WOOH formation on the redox states of PKG1 α , Prx2 and Prx4 in mesenteric resistance arteries taken from LPS-treated mice after one-minute infusion of vehicle or Trp (8 mM) via the jugular vein. b-d) Summary data of experiments performed in a), Trp induced significant dimerization of PKG1 α (b), without altering the thiol redox state of Prx2 (c) or Prx4 (d). Data are representative of six independent experiments and shown as mean \pm SEM (b-d). Statistical analysis was performed using two-tailed Mann-Whitney tests with * $p \leq 0.05$.

Overall, we consider the strengths of these data in support of the physiological importance of our findings to be greater than can be achieved with experiments using isolated arteries.

In the current submission, we used AUR, BCNU and H₂O₂ as *tools* to indicate that there may be a change in arterial redox pathways in response to IDO1 expression following treatment of the animals with LPS. We are aware of potential issues of L-NAME to interfere with the IDO1/*cis*-WOOH/PKG1 α pathway. For instance, in the absence of L-NAME, production of nitric oxide may inhibit *cis*-WOOH formation (*J Biol Chem* 1994;269:14457; *ibid* 2007;282:23778; *Nature* 2019;566:548) or the presence of L-NAME may enhance *cis*-WOOH formation by increasing H₂O₂ (*Nature* 2019;566:548). However, the precise details of these relationships are beyond the scope of the submitted manuscript. We instead rely on our *in vivo* data (**Figures 1-2**) which demonstrate that the IDO1/*cis*-WOOH/PKG1 α pathway is (patho)physiologically relevant and associated with the apparent selective oxidative dimerization of PKG1 α but not Prx2/4.

To address the reviewer's comments has required additional animals as arteries need to be used within hours of taking them from the animal. We believe that such experiments to the fullness of the reviewer's request are hard to justify from an ethical point of view. However, as proof-of-principle, we have repeated experiments examining relaxation of resistance arteries from naïve mice to AUR in the absence and presence of L-NAME plus indomethacin. Arteries from naïve mice were chosen because their relaxation was substantially greater than that seen with arteries from LPS-treated mice, so that any effect of L-NAME/indomethacin should be particularly apparent. The results obtained show that AUR induces comparable relaxation in the absence and presence of both inhibitors (**Figure 3**). This finding, added to Supplementary Information (**Fig S1c**) of the revised manuscript, suggests that AUR-induced relaxation occurs in the presence of active eNOS and COX, consistent with the *in vivo* data showing that peroxide-mediated arterial relaxation takes place in the presence of active eNOS and COX (**Figures 1-2**).

Figure 3. Effect of L-NAME and indomethacin on auranofin-induced arterial relaxation. Arterial relaxation responses to 300 nM auranofin in third order mesenteric arteries from naïve mice in the presence of Krebs buffer (grey circles n=4) compared to arteries bathed in Krebs containing 100 μ M L-NAME and 10 μ M indomethacin (black circles n=4).

2) LPS induces increases in some of the monomers (like Prx2 in RBCs). This might easily be a consequence of overoxidation. Is this so? This is an important aspect as dimer formation is obviously not the only sign of oxidation.

Response: The reviewer raises a valid methodological point. Hyperoxidized Prx2 can appear as monomer. With H₂O₂ as the oxidant, hyperoxidized Prx2 monomer is typically formed at high molar ratios of oxidant to protein (*J Biol Chem* 2013;288:14170). However, in our original manuscript, we did examine whether exposure to *cis*- and *trans*-WOOH can give rise to hyperoxidized Prx2/3 monomer using an antibody that recognizes Prx-SO_x. We documented (bottom panels in Figure 3e of original manuscript) that under the experimental conditions used, hyperoxidation occurs with H₂O₂ but **not** with *cis*-WOOH, even at high molar ratios of hydroperoxide to Prx2 (shown here as **Figure 4**) and Prx3 (not shown). In response to the reviewer, we have changed the subheading to the legend of Figure 4e in the revised manuscript from “Hyperoxidation ...” to “Absence of hyperoxidation of His-tagged Prx2 (20 μ M) upon reaction with *cis*-WOOH”.

Figure 4. Absence of hyperoxidation during the reaction of His-tagged Prx2 (20 μ M) with *cis*-WOOH. Samples were assessed in duplicates using non-reducing SDS-PAGE, with proteins on one gel silver stained (upper panels) and the other transferred to a polyvinylidene difluoride membrane before probing with an antibody specific for Prx-SO_{2/3} (lower panels). Data shown are representative of two separate experiments.

We are not aware of literature evidence for LPS increasing RBC Prx2 *monomer in vivo*. What has been reported previously (*FASEB J* 2013;27:3315-3322) is that LPS temporarily increases Prx2 dimer in blood cells, with maximal oxidation observed 10 h after LPS administration. In response to the Reviewer, we have examined the effect of LPS on the redox state of Prx2 in blood cells (predominantly RBC) and arteries collected 16 h post LPS administration, *i.e.*, when IDO activity is maximal, and the same time point used in the submitted manuscript. NEM was immediately added to block *ex vivo* Prx2 oxidation, and samples were added to non-reducing gel loading buffer and subjected to SDS-PAGE followed by immunoblotting for Prx2. At this time point, LPS had no measurable effect on the thiol redox state (monomer or dimer to total protein) of Prx2 in RBC and arteries (**Figure 5**). This information has been added to **Figure 1i** of the revised manuscript.

Figure 5. Prx2 expression and thiol oxidation state is unchanged in response to LPS. Wild-type C57BL/6J mice were injected with LPS and blood samples were collected at 16 hours into maleimide (100 mM final concentration) to alkylate free thiols. Mice were then perfused with PBS containing NEM to alkylate arterial tissue and abdominal and mesenteric arteries were isolated, snap frozen and powdered. Non-reducing gel loading buffer was added to RBC and powdered arterial tissue and all samples were centrifuged at 21,000 x g for 10 min. Supernatants were subjected to SDS-PAGE followed by immunoblotting for Prx2. There is no evidence that LPS alters the thiol redox state of Prx2 in RBC or either arterial bed tested. The high level of reduction of RBC Prx2 suggests that inadvertent *ex vivo* oxidation of Prx2 was minimal. Data are shown as individual data point with mean.

3) Given that WOOH and H2O2 both act through PKG oxidation, it is hard to understand that LPS, which is used to induce IDO-1 and to therefore to produce WOOH, does not impact on the H2O2-induced relaxation. In my opinion, this contradicts the authors' hypothesis.

Response: The reviewer raises an interesting point. The data likely referred to by the Reviewer (Figure 1g) are results obtained with *exogenously* added bolus H₂O₂. The purpose of that experiment was to ascertain that LPS treatment does not change relaxation mediated by *bolus* H₂O₂, indicating that LPS does not cause substantial changes to either endogenous defence systems against H₂O₂ and oxidant-induced activation of PGK1 α . Importantly, we do not imply that relaxation by *bolus* H₂O₂ is representative of that induced by endogenously formed H₂O₂.

Previous studies using the redox-dead PGK1 α mutant mouse have provided strong evidence for a role of an endogenous oxidant, commonly believed to be H₂O₂, in mediating ACh-induced relaxation of mouse mesenteric arteries (*Nature Med* 2012;18:286). We already demonstrated that LPS indeed affects such relaxation in an IDO1-dependent manner (Figure 4g, h in *Nature* 2019;566:548; reproduced as **Figure 6** below). Thus, in the absence of Trp, ACh-induced relaxation of isolated arteries from LPS-treated mice is decreased compared with naive controls (R_{max} : 67.6% LPS mice vs 83% naive mice) (**Figure 6**). Addition of L-Trp to arteries from LPS-treated mice increased ACh relaxation to values comparable to that seen in arteries from naive mice in the absence or presence of Trp (R_{max} relaxation: 88.3% LPS mice + Trp vs 83% naïve control and naïve mice + Trp 76.2%), whereas addition of the IDO1 inhibitor 1-methyl-tryptophan (1-MT) decreased ACh-induced relaxation in LPS treated mice (R_{max} relaxation: 67.6% LPS mice control vs 44.7% LPS mice + 1-MT) (**Figure**

6). We therefore pose that IDO1 expression does modulate arterial relaxation to endogenously formed H_2O_2 as shown in our previous data but these somewhat subtle changes to ACh-induced relaxation may not be apparent when relaxation is induced by high concentrations of *bolus* H_2O_2 .

Figure 6. Regulation of arterial relaxation by IDO1-mediated L-Trp (Trp) metabolism. Acetylcholine (ACh)-induced relaxation of mouse mesenteric resistance arteries from naïve (left) or LPS-treated (right) wildtype mice in the absence (control, n=4 for naïve, n=13 for LPS) or presence of Trp (n=4 for naïve, n=10 for LPS) or 1-methyl-tryptophan (1-MT, n=4 for naïve, n=8 for LPS). Data are mean \pm SEM. * $p \leq 0.05$ as determined by 1-way ANOVA of maximal relaxation values with Dunnett's test. Reproduced from *Nature* 2019;566:548.

It is difficult to deduce the precise contribution of H_2O_2 versus *cis*-WOOH to arterial relaxation following LPS treatment, because LPS has numerous biological effects on arteries. For example, we previously reported (*Nature* 2019;566:548) that in addition to inducing arterial IDO1 expression, LPS increases endogenous H_2O_2 in arteries and that this results in increased formation of *cis*-WOOH (**Figure 7**) and PKG1 α oxidation in the presence of added Trp (not shown). For these reasons, there is *a priori* no contradiction in our observations because IDO1 requires H_2O_2 to form *cis*-WOOH (**Figure 8**).

Figure 7. Arterial formation of *cis*-WOOH. Left, Endogenous formation of H_2O_2 by porcine arteries in the absence (-) and presence of 400 ng mL⁻¹ recombinant porcine interferon- γ (rpIFN γ) (+). Right, Endogenous formation of ¹⁵N₂ *cis*-WOOH from ¹⁵N₂ L-Trp by porcine arteries pretreated with 0, 200 or 400 ng mL⁻¹ rpIFN γ . Data are representative of 5 (left) and 6 (right) independent experiments. * $p \leq 0.05$ using Mann-Whitney rank sum (left) or Kruskal-Wallis with Dunn's post-test (right). Reproduced from *Nature* 2019;566:548.

Figure 8. Formation of *cis*-WOOH via oxidative activation of IDO1 dioxygenase activity requires H_2O_2 . a, Total LC-MS/MS ion chromatogram of reaction mixtures of recombinant IDO1 and Trp under reducing (top, middle) and oxidizing conditions (H_2O_2 , bottom). b, IDO1-mediated formation of Trp-derived metabolites with successive addition of H_2O_2 . The chemical structure of the peroxidase product, oxyindolylalanine (OIA), is shown in (b). Data are representative of 3 independent experiments (a) or mean \pm SEM of 3 (b), independent experiments. Reproduced from *Nature* 2019;566:548.

4) On page 7 line 151 the authors aim to estimate the GSH concentration in arteries, but the calculation appears very shaky, given that the cytosolic GSH content is what matters. Vessels have a large extracellular space and a high connective tissue content. Given that the value is surprisingly low and that some further argumentation depends on this value, the authors should seek a way to accurately measure GSH cytosolic content (or better said the cytosolic volume).

Response: The issue raised by the reviewer is valid. The GSH concentration determined does not represent a cellular concentration, as arteries have an extracellular space and a connective tissue basement membrane. The revised manuscript now refers to a previous study documenting the relative contribution of cellular and non-cellular compartments to arteries, according to which endothelial and smooth muscle cells account for 74% of resistant arteries (*Blood Vessels* 1983;20:72-91). We have

clarified the corresponding text and adjusted further argumentation that depends on this value, although we believe that our overall conclusion is not altered substantially.

5) On page 15, line 330/331 the authors suggest that the relaxation to H₂O₂ is controlled by GPx and Prxs, but this information is missing in the result section. Do BCNU and AUR indeed block the H₂O₂-induced relaxation?

Response: (see also our response to Query 3 of Reviewer 3). The corresponding text reads “our results from arteries isolated from naïve mice indicate that this relaxation is *regulated* by arterial GPx and Prxs”. We base this statement on the data in Figure 1d showing that BCNU and AUR induce relaxation and propose that this is through ‘releasing the brake’ on the signalling by endogenous H₂O₂ (Figure 1b). It follows that BCNU and AUR would be expected to potentiate (not “block”) relaxation induced by H₂O₂ as the two compounds inhibit the enzymatic removal/metabolism of H₂O₂.

6) What is the effect of auranofin on vascular tone in the PKG1 α C42S mice? This aspect is missing in the figures.

Response: We apologize for this omission in the original manuscript. In the revised manuscript, we now show that auranofin-induced relaxation of resistance arteries is decreased in naïve PKG1 α C42S mice (**Figure 9**), consistent with such relaxation being mediated by H₂O₂. This information has been added as Figure 1f (left hand panel) of the revised manuscript.

Figure 9. Effect of auranofin on arterial relaxation in WT and PKG1 α C42S mice. Arterial relaxation responses to 300 nM auranofin in mesenteric arteries from WT and PKG1 α C42S redox dead knock-in (KI) C57Bl/6N mice (n=7-8) * p<0.05 (Mann-Whitney).

Reviewer #2 (Remarks to the Author):

In their manuscript “Stereospecific impediment of reductive inactivation of tryptophan-derived hydroperoxide: a new model for mammalian redox signaling by hydrogen peroxide” Queiroz et al present a novel and promising H₂O₂-dependent mechanism of redox signaling. The proposed model goes like this: H₂O₂ is enzymatically transformed into a tryptophan-derived hydroperoxide (WOOH), which in turn oxidizes the final target: protein kinase G1 α . The intermediate hydroperoxide would then “protect” the oxidant from the main peroxidases Prx and GPx. An idea akin to the redox relay model but with a hydroperoxide instead of sulfenic acid/disulfide intermediate.

The authors study in detail the reaction of human Prx2 and GPx4 with two epimers of WOOH, use an ex-vivo model of mouse artery relaxation combined with inhibition of the Trx and GSH systems to identify the involvement of Prx and GPx in interfering the dimerization of PKG1 α .

Although the idea is very interesting, there are a few important questions that need to be addressed before the manuscript is published:

Response: We appreciate the detailed and constructive comments as well as the overall positive assessment of our work by this reviewer.

1) Spectroscopic properties of WOOHs. *A few characteristics of the reduction products (WOH) are mentioned, such as extinction coefficient and their UV spectra but I could not find any data on the WOOHs. The question is important because a big portion of the kinetic study is made through the intrinsic fluorescence of Prx2. Questions that need to be answered are:*

Response: We apologize for the omission of extinction coefficients for *cis*- and *trans*-WO(O)H. The detailed characterization of these compounds is part of a separate, manuscript recently published in *Nature Protocols*. This reference has been added to the revised manuscript.

a. Are *cis* and *trans* WOOH fluorescent?

Response: *cis*- and *trans*-WOOH are weakly fluorescent, as published recently (*Nat Protoc* 2021 16:3382-418). Specifically, both hydroperoxides exhibit ~1% of the fluorescence of equimolar Trp. By comparison, the alcohols are somewhat more fluorescent: *cis*-WOH and *trans*-WOH exhibit 30% and 16% fluorescence, respectively, of an equimolar solution of Trp (**Figure 10**).

Figure 10. Fluorescent properties of Trp, *cis*- and *trans*-WO(O)H determined in isolation using a fluorimeter and wavelength settings used for the stopped-flow experiments, *i.e.*, Ex_{280 nm}, Em_{>320 nm}.

b. Do they interfere with the detection of Prx2 fluorescent changes, through FRET, quenching or inner filter effect?

Response:

We thank the Reviewer for raising this important issue. To assess whether the presence of *cis*-WO(O)H affects Prx2 fluorescence, we used C51S His-tagged Prx2 (2 μ M) and *cis*-WO(O)H at various molar ratios used for the stopped flow conditions in Fig. S2 of the manuscript. Increasing *cis*-WOOH in the presence of a constant (and maximally expected concentration of *cis*-WOH, *i.e.*, 2 μ M) *decreased* fluorescence in a concentration-dependent manner (**Figure 11**). The extent of this *cis*-WOOH-mediated fluorescence decrease was greater than that reported in Fig. S2 of the original manuscript, indicating that the fluorescence-based stopped flow experiments to determine a rate constant for the reaction of Prx2 with *cis*-WOOH do not represent a valid approach. The revised manuscript now mentions that we tried to use stopped flow experiments but the contribution of the fluorescence to WOOH and WOH interfered, so that it was not possible. Data from stopped flow experiments (Figs. 2c-l, Tables S1 and S2, Fig. S2) have been deleted from the revised version of the manuscript.

Figure 11. Assessment of potential interference of *cis*-WOOH with endogenous fluorescence changes induced by Prx2 cysteine oxidation and assessed by stopped-flow fluorescence. Fluorescence of 2 μ M His-tagged C51S Prx2 was determined in the stopped flow spectrometer in the presence of 2 μ M *cis*-WOH and increasing concentrations of *cis*-WOOH using Ex_{280 nm} and Em_{>320 nm}. The increasing concentrations of *cis*-WOOH used mimicked those used in Fig. S2B or the original manuscript. Addition of *cis*-WOOH decreased endogenous fluorescence of his-tagged C51S Prx2 in a concentration-dependent, bi-phasic manner.

In light of these problems associated with the stopped-flow experiments, we carried out a competition experiments to support the redox blotting approach in the original manuscript (Supplementary Figure S3). We reacted *cis*- and *trans*-WOOH with thioredoxin reductase (TrxR) \pm increasing amounts of Prx2 and followed NADPH consumption as a decrease in 340 nm absorbance. TrxR is known to reduce lipid and organic hydroperoxides (*J Biol Chem* 1995;270:11761; *Arch Biochem Biophys* 1999;363:19). The rate of reaction between TrxR and *cis*- and *trans*-WOOH was determined as 2.1×10^4 and $1.8 \times 10^4 \text{ M}^{-1}\text{s}^{-1}$, respectively, using the initial rates of NADPH oxidation (**Figure 12, left**). The rate of NADPH oxidation observed were linear for approximately the initial 30 s of the reaction

and were significantly greater than those previously reported for lipid hydroperoxides, *t*-butyl- and cumene hydroperoxide (*J Biol Chem* 1995;270:11761; *Arch Biochem Biophys* 1999;363:19). Data from the competition experiment (**Figure 12, right**) suggest that the rate of reaction of Prx2 with *cis*- and *trans*-WOOH is equal or less than that for the reaction between the hydroperoxides and TrxR, *i.e.*, $\leq 10^4 \text{ M}^{-1}\text{s}^{-1}$. This value is in good agreement with both, the rate constants determined from redox blotting, *i.e.*, 3.8×10^3 and $1.8 \times 10^4 \text{ M}^{-1}\text{s}^{-1}$ for *cis*- and *trans*-WOOH, respectively, as well as rate constants for the reaction of amino acid hydroperoxides with Prx2 (*Biochem J* 2010;432:313-321). Importantly, these data support the conclusion from our original studies, *i.e.*, the Prx2 reacts more slowly with *cis*-WOOH compared with H_2O_2 , thereby facilitating redox signaling by *cis*-WOOH. This new information is added to **Figure S6a-b** of the revised manuscript.

Figure 12. Determination of the rate of reaction between *cis*- and *trans*-WOOH and Prx2 using competition kinetics with thioredoxin reductase (TrxR). Left, *cis*-WOOH (filled circles) or *trans*-WOOH (open circles; 10–100 μM) were incubated with 0.2 μM TrxR and 200 μM NADPH. NADPH consumption at 340 nm was measured over the initial 30 s at 25 °C using a spectrophotometer. NADPH consumption versus *cis*-WOOH and *trans*-WOOH concentrations was plotted to derive the rate of reaction between TrxR and *cis*-WOOH ($k = 2.1 \times 10^4 \text{ M}^{-1} \text{ s}^{-1}$) and *trans*-WOOH ($k = 1.8 \times 10^4 \text{ M}^{-1} \text{ s}^{-1}$). Right, pre-reduced purified Prx2 (1–20 μM) was incubated with 10 μM *cis*-WOOH (filled circles) or *trans*-WOOH (open circles), 0.2 μM TrxR and 200 μM NADPH at 25 °C. NADPH consumption at 340 nm was measured over 400 s using a spectrophotometer.

The above results were validated two-fold. First, we confirmed by Coomassie staining that the Prx2 used was initially reduced (**Figure 13**). Second, we confirmed that reaction with TrxR resulted in the conversion of *cis*- and *trans*-WOOH to *cis*- and *trans*-WOH, respectively (**Figure 14**). The alcohols were the only products detected and accounted for 62 and 84% of the loss of *cis*- and *trans*-WOOH, respectively. These results have been added as Supplementary Information (**Fig S6c-d**) to the revised manuscript.

Figure 13. Determination of redox status of Prx2 (10 μM) during before (0 and 3 min) after (10 min) reaction with 10 μM *cis*- or *trans*-WOOH, TrxR (0.2 μM) and NADPH (200 μM).

Figure 14. Determination of products by LC-MS/MS after 10 min reaction of 50 μM *cis*- (left) or *trans*-WOOH (right) with TrxR (0.2 μM) and NADPH (200 μM) at 25 °C.

c. The same questions for the products *cis*- and *trans*-WOH.

Response: See above.

On the same topic of spectroscopic properties, the results on Fig 4a and 4b are hard to understand, the WOH used as ligands have their own spectra, what is shown in 4a is in any way different from the sum of the spectra of protein + WOH? What is meant by the spectral shift? I am sure the authors know that the chromophore of Prx2 is a combination of its 2 Trp and its 7 Tyr residues, then, what is shifting? Can the authors show a differential spectrum? Can the binding equilibrium be followed through absorbance measurements instead of rather complex MS experiment as presented?

Response: We apologize for the lack of clarity. The purpose of Figure 4a (**Figure 5a in the revised manuscript**) is to demonstrate that the absorbance maximum of *cis*-WOH (293 nm) and *trans*-WOH (292 nm) “shifts” to 284 and 285 nm, respectively, upon addition of Prx2, indicative of binding of WOH to the protein. This has been clarified in the Figure legend of the revised manuscript.

As suggested by the Reviewer, we did attempt to determine the binding equilibrium of WOH with Prx2 through absorbance measurements. However, the comparatively strong absorption of Prx2 at 277 nm combined with its proximity to the absorption maxima of WOH and the low extinction coefficients of WOH, does not allow for such determination to be accurate. It is for this reason that we employed native mass spectrometry for binding equilibrium measurements.

We respectfully point out that native mass spectrometry is a straightforward approach to directly and reliably determine ligand-protein binding constants. In the main text, we now reference what we consider to be the two best reviews in the literature of the native MS approach for quantifying ligand-protein binding constants to highlight the reliability and simplicity of this approach (*Int J Mass Spectrom* 2002;216:1; *Curr Proteom* 2011;8:47). These reviews are from the laboratories of Prof Renato Zenobi (ETH Zürich) and Prof Carol Robinson (Oxford). We have also added a sentence with an additional reference (*Anal Chem* 2020;92:4614) to further highlight the reliability of the K_d measurements in the main text.

2) Prx1. The authors could not find a suitable antibody for assessing the presence of Prx1 and then they did not study its potential involvement in the redox signaling. This represents a major weakness in the manuscript, since Prx1 is the main cytosolic peroxiredoxin in most human cell types (Selvaggio, G. et al *Redox Biol.* 15, 297–315 (2018)).

Response: The reviewer raises a limitation of our original submission, *i.e.*, it did not assess the potential role of Prx1 in redox signalling by *cis*-WOH. We have since obtained an anti-human Prx1 antibody that detects mouse Prx1 and used it to estimate the concentration of Prx1 in mouse abdominal and mesenteric arteries. Prx1 was assessed to be present at <0.5 fmol/μg protein in arteries from naïve mice (**Figure 15, top**) and at ~3 fmol/μg protein in arteries from LPS-treated mice (**Figure 15, bottom**), using recombinant human Prx1 for standardization. We assumed comparable reactivity of the antibody with mouse and human Prx1, given the 95% homology in amino acid sequences. This new information has been added to Table 1 and Figure S2 of the revised manuscript.

Figure 15. Expression of Prx1 in mouse arteries. Immunoblotting was used to quantify the expression of Prx1 in abdominal aorta (A) and mesenteric (M) arteries from naïve (n=3, top) and LPS-treated mice (n=3, bottom). Expression of endogenous Prx1 was determined against a standard curve of recombinant (r) human Prx1 using 15-25 μg of total arterial homogenate.

These additional results indicate that arteries contain less Prx1 than Prx2. Also, Prx1 reacts with organic hydroperoxides about 5-times slower than Prx2 (*J Biol Chem* 2017;292:8705). A bacterial Prx1/alkyl hydroperoxide reductase C (AhpC) also reacts ~180-times slower with bulkier hydroperoxides than H₂O₂ (*Biochemistry* 2015;54:1567). Together, these results support our focus of the kinetic experiments on Prx2. This information has been added to the revised manuscript (page 9).

The reviewer’s assertion that Prx1 “is the main cytosolic peroxiredoxin in most human cell types” is based on the report by Selvaggio *et al.* (*Redox Biol* 2018;15:297). That paper used literature values for the contents of cytosolic Prx1, 2 and 6 in thirteen human cell types for *modelling* of

experimental data without including information on reproducibility or variation. Most of the cell types used were cancer cell lines, the proteome of which can differ from that of primary cells (*J Proteomics* 2016;136:234). Also, unpublished data from one of us (CCW) indicates that the relative levels of Prx1 and 2 vary considerably between different situations, so that the data reported by Selvaggio *et al.* (*Redox Biol* 2018;15:297) is difficult to extrapolate to *primary* arterial cells. In any case, our data indicates that Prx2 predominates in resistance arteries and is the most likely regulator of *cis*-WOOH induced relaxation.

3) The kinetic competing systems. According to the model, to oxidize PKG1 α , H₂O₂ is first transformed into WOOH by IDO1, therefore, there are two important kinetic systems in competition.

a. Firstly, IDO1 competes with the peroxidases (Prx and GPx) for the reduction of H₂O₂. No mention of the reaction of the kinetics of IDO1 as a peroxidase is made in the manuscript nor I could find easily a rate constant that indicates the likelihood of H₂O₂ ending as a substrate of IDO1 instead of Prx and GPx, to support their claim the authors should show that the reaction of IDO1 is at least competitive.

Response: We agree with the reviewer that for production of *cis*-WOOH, IDO1 has to be able to effectively compete with Prx and GPx for H₂O₂. Indeed, we reported previously that incubation of ¹⁵N₂-Trp with IDO1-expressing porcine arteries resulted in the accumulation of ¹⁵N₂ *cis*-WOOH (see **Figure 7** above). This implies that in intact arteries IDO1 can effectively “compete” for H₂O₂ with other targets including GPxs, Prxs and PKG1 α .

Others previously reported on the kinetics of the reaction of IDO1 with H₂O₂ (*J Biol Chem* 2011;286:21220). Using time-resolved optical absorption changes, Lu and Yeh reported an apparent bimolecular rate constant of $8 \times 10^3 \text{ M}^{-1} \text{ s}^{-1}$ for the formation of the IDO1 compound II type ferryl species (Fe⁴⁺=O²⁻). It is noteworthy that this approach does not detect the initial ferric-(hydro)peroxo intermediate (Fe³⁺=O(H)OH) and hence does not directly determine the kinetics of the initial reaction of IDO1 with H₂O₂. We also note that the reaction of heme peroxidases, such as myeloperoxidase (*Biochim Biophys Acta* 1988;955:337-345) and horseradish peroxidase (*J Biol Chem* 1987;262:2576-2581) with H₂O₂ to form the respective compound I is faster than formation of compound II.

To address the Reviewer’s concern, we carried out experiments where IDO1 competes with Prx2 for H₂O₂ in the presence of L-Trp and simultaneously measured formation of *cis*-WOOH and oxyindolylalanine (OIA) by LC-MS/MS. These products were chosen as they are the specific reaction products of IDO1 with H₂O₂ via its oxidative dioxygenase or peroxidase activity, respectively. Initially we added 2 μM H₂O₂ to solutions of L-Trp containing 2 μM IDO1 in the absence or presence of 4 μM fully reduced His-tagged wild type human Prx2 (2:1, Prx2:IDO1). Under these conditions the presence of Prx2 led to significantly decreased but still detectable *cis*-WOOH/OIA formation (**Figure 16, left**). We then repeated the experiment using IDO1 in 5-fold molar excess of Prx2, a biologically feasible scenario in conditions of LPS-induced inflammation. Once again, formation of *cis*-WOOH/OIA was decreased but still detectable in the presence of Prx2 compared to control (12% of control; **Figure 16, left**). Using this data and the rate constant for the reaction of Prx2 with H₂O₂ ($k = 2 \times 10^7 \text{ M}^{-1} \text{ s}^{-1}$), we estimated the rate constant for the reaction of IDO with H₂O₂ in the presence of L-Trp to be $\sim 3.2 \times 10^5 \text{ M}^{-1} \text{ s}^{-1}$. We then carried out *in silico* analyses using this rate constant or the rate determined by Lu and Yeh and applied them to the experimental data in the LC-MS/MS experiments (20 μM IDO1). The simulation predicted *cis*-WOOH/OIA formation and an associated decrease in Prx2 dimerization when the rate of reaction between IDO1 and H₂O₂ was set at $10^5 \text{ M}^{-1} \text{ s}^{-1}$ (**Figure 16, top right**) but not $10^3 \text{ M}^{-1} \text{ s}^{-1}$ (**Figure 16, bottom right**).

Figure 16. IDO+H₂O₂-mediated oxidation of L-Trp in the absence or presence of 4 µM Prx2. Left, rIDO1 (2 or 20 µM) ± pre-reduced His-tagged WT Prx2 (4 µM) was incubated with L-Trp (100 µM) and H₂O₂ (2 µM) for 5 min at 25 °C before *cis*-WOOH and oxyindolylalanine were determined by LC-MS/MS as described previously (*Nature* 2019;566:548). Data represents mean ± SEM (n=4). Right, *in silico* simulation of Trp oxidation products using 4 µM Prx2, 20 µM IDO1, 2 µM H₂O₂ and a rate constant of 3.2 × 10³ M⁻¹s⁻¹ (top) or 8.0 × 10³ M⁻¹ s⁻¹ (bottom) for the reaction of IDO1 with H₂O₂. Simulation was obtained using Gepasi software.

Following LC-MS/MS analyses, samples were alkylated with NEM and probed by non-reducing SDS-PAGE and Western blotting for Prx2 monomer and dimer (**Figure 17, top**) and hyperoxidized Prx2 (**Figure 17, bottom**). This confirmed that under the reaction conditions employed (IDO1+Prx2+L-Trp+H₂O₂; 5 min; 25°C): i) catalytically active Prx2 monomer remained present at the end of the reaction (Lanes 3 and 6); and ii) hyperoxidized Prx2 monomer was absent (lanes 3 and 6). Higher concentrations of IDO1 lead to greater amounts of Prx2 monomer at the end of the 5 min reaction (lane 6 vs lane 3). Together, these results indicate that IDO1 can utilize H₂O₂ for

Figure 17. Prx2 (4 µM) is not completely dimerized or hyperoxidized in the presence of IDO1 (2 or 20 µM) and H₂O₂ (2 µM). Upper, anti-Prx2 Western blot of samples run on the LC-MS/MS (samples alkylated with NEM post-LC-MS/MS analyses). Lower, anti-Prx-SO₃ Western blot of samples run on the LC-MS/MS (samples alkylated with NEM post-LC-MS/MS analyses).

enzymatic oxidation of L-Trp to *cis*-WOOH in the presence of Prx2. The data shown in Figures 16-17 has been added to the revised manuscript as **Figure 6a-b** and **Fig S9**.

b. Secondly, the peroxidases Prx and GPx compete with PKG1α for the reduction of WOOH. The authors dedicate a large fraction of the manuscript to describe the kinetics of *cis* and *trans* WOOH with Prx2 and GPx4 which they identify as the major peroxidases, but no parallel study is made on PKG1α. Then, although WOOH appears to be a rather limited substrate, particularly for Prx2, it is not clear how fast is the reaction with the cysteine of PKG1α.

Response: We previously reported (*Nature* 2019;566:548) on the ability of *cis*- and *trans*-WOOH to oxidise cysteine residue 42 in PKG1α. Specifically, we observed such oxidation to be more pronounced with *cis*-WOOH compared with *trans*-WOOH (**Figure 18**). These results indicate stereospecific oxidation of PKG1α by WOOHs. This situation is similar to that with Prx2/3, except that compared with each other, *cis*-WOOH favours oxidation of PKG1α whereas *trans*-WOOH favours oxidation of Prx2/3. Overall, this facilitates redox signalling of *cis*-WOOH to PKG1α.

Figure 18. Oxidation of cysteine residue 42 of PKG1 α by *cis*- but not *trans*-WOOH. Dimerization of recombinant human wildtype and cysteine 42 serine (C42S) mutant PKG1 α by equal concentration of *cis*- or *trans*-WOOH \pm light and the transition metal chelator diethylenetriaminepentaacetic acid (DTPA); n=3 separate experiments. Compared with control (Ctrl), *trans*-WOOH did not cause significant dimerization. Reproduced from *Nature* 2019;566:548.

We agree with the reviewer that data on the kinetic of the reaction of *cis*-WOOH with PKG1 α would provide additional useful information. We therefore compared the extent of PKG1 α dimerization by *cis*-WOOH and H₂O₂ over time. As Cys42 in PKG1 α is highly sensitive to autoxidation, the thiol reducing agent TCEP was added in molar excess over PKG1 α , as described previously by others (*J Biol Chem* 2018;293:16791–16802). Under these conditions, *cis*-WOOH caused greater PKG1 α dimerization than the same concentrations of H₂O₂ (**Figure 19**). While exact rate constants cannot be extrapolated from this data, it suggests that the reaction between *cis*-WOOH and PKG1 α is kinetically more favourable than the reaction of PKG1 α with H₂O₂. We have added this information as **Figure 7e-f** of the revised manuscript.

Figure 19. Time-dependent oxidation of PKG1 α by *cis*-WOOH or H₂O₂. PKG1 α (3.3 μ M in phosphate buffer (25 mM) pH 7.0 containing TCEP (85 μ M) was incubated at 25 °C with 200 μ M *cis*-WOOH or H₂O₂ for the indicated time before maleimide (125 mM) was added to stop the reaction. Samples were subjected to SDS-PAGE followed by Coomassie staining (A; top left) and quantification of the results (B; top right). The results shown are from a single experiment, representative of three separate experiments.

4) Some serious problems in the kinetics as presented.

a. The pseudo first order rate constants of GSH with H₂O₂ (and presumably the WOOHs) are overestimated, by dividing by [GSH] they yield second order rate constants above 100 M⁻¹ s⁻¹. One of the authors published a seminal paper reporting that constant as 0.87 M⁻¹ s⁻¹ by a much robust method (Winterbourn, C. C. & Metodiewa, D. *Free Radic. Biol. Med.* 27, 322–328 (1999)).

Response: We modified “Determination of the rate constant for the reaction of GPx4 with *cis*-WOOH and *trans*-WOOH” in the methods section as follows:

“Absorbance data at 340 nm from the progression curves of NADPH oxidation, obtained in the presence of GSH and triggered by 2 x 10⁻⁵ M peroxidic substrate, were used to calculate pseudo first order rate constants. Peroxidic substrate concentrations at each interval time (1 s) were calculated using an extinction coefficient of 6,220 M⁻¹cm⁻¹. The slope (k’) of the straight line obtained by plotting ln[A] vs time (s) is k[B]₀ and represents the pseudo first order rate constant, where [A] is the concentration of the peroxidic substrate and [B]₀ is the concentration of GSH kept constant by GSSG reductase activity. Second order rate constants for reaction of the peroxidic substrates with GSH were calculated by plotting pseudo first order rate constants against the GSH concentration.

We also changed Figure 2a as follows:

Second order rate constants (M ⁻¹ s ⁻¹) - GPx4		
	k_{+1}	k_{+2}
PC-OOH	8.34 x 10 ⁷	2.94 x 10 ⁴

H ₂ O ₂	4.78 x 10 ⁵	7.98 x 10 ⁴
cis -WOOH	1.02 x 10 ⁶	1.80 x 10 ⁴
trans -WOOH	1.59 x 10 ⁶	2.46 x 10 ⁴

Second order rate constant (M⁻¹s⁻¹) – GSH	
H ₂ O ₂	0.49
cis -WOOH	5.91
trans -WOOH	7.93

Finally, we changed the legend to Figure 2a as follows:

“Figure 2. Reactions of *cis*- and *trans*-WOOH with GPx4 and GSH. c) Upper panel shows the second order rate constants for the reaction of *cis*- and *trans*-WOOH with GPx4. k_{+1} is the rate constant for the reduction of the hydroperoxide, k_{+2} is the cumulative rate constant for the regeneration of the reduced enzyme. The lower panel shows second order rate constants for the non-enzymatic reaction of *cis*- and *trans*-WOOH with GSH. H₂O₂ and phosphatidylcholine hydroperoxide (PC-OOH) were used for comparison.”

b. Page 8, Line 176, and this is a red flag: “...and the dissociation constant (k-1) as -3.1 ± 0.8 s-1”. There are NO negative rate constants (or dissociation constants) and dissociation constants have units of concentration, not frequency.

Response: The reviewer is correct. There are no negative rate constants. As discussed above (see response to Reviewer #2, 1b), stopped-flow experiments following changes in Prx2 intrinsic fluorescence cannot be used reliably to determine the rate of reaction between *cis*-WOOH or *trans*-WOOH with Prx2 because of fluorescence interference from both the hydroperoxide and the corresponding hydroxides (see **Figure 11**). Due to this, rate constant data derived from stopped-flow experiments have been removed from the revised manuscript.

c. Figure 2d and S2b, the broken line profile of the kobs vs [WOOH] plot deserves an explanation, it is unexpected and hard to reconcile with a simple reaction scheme as proposed. What exactly happens at that 85 micromolar WOOH? Why the change in reaction order from zero to one?

Response: As discussed above (see response to Reviewer #2, 1b), stopped-flow experiments following changes in Prx2 intrinsic fluorescence cannot be used reliably to determine the rate of reaction between *cis*-WOOH or *trans*-WOOH with Prx2 because of fluorescence interference from both the hydroperoxide and the corresponding hydroxides (see **Figure 11**). Due to this, rate constant data derived from stopped-flow experiments have been removed from the revised manuscript.

d. The Km for GSH with GPx is “indefinite” according one of the authors (Ursini, F., et al BBA - Gen. Subj. 839, 62–70 (1985), so how can it be GSH 4 fold lower than the Km?

Response: The reviewer is correct in that the K_m of GPx for GSH is indeed indefinite, as shown by Flohé and Ursini. In this context, it was inappropriate to cite ref. 29. For this reason, Ref. 29 has been deleted from the revised manuscript.

The comment related to the arterial concentration of GSH was meant to refer to the two GSH-dependent reactions in the reductive part of the catalytic cycle of GPx, *i.e.*, the formation of the selenosulfide from the selenenic acid, and the subsequent reduction of the selenosulfide. In the revised manuscript, we have clarified the relevant text on **page 8** follows: “Given that the GSH-dependent reactions of the reductive part of the cycle are rate-limiting ($k_2 \ll k_1$; see Fig. **1c in the revised manuscript**), the low arterial concentrations of GSH can reasonably be expected to slow down the overall catalytic activity of GPx4?”

e. It is not clear what functions the authors used to fit their experimental time courses, it says in p. 8 l 17 “the initial rapid decay fitted a single exponential equation” but clearly the initial decay cannot be separated from the slower increase so a sum of at least two exponential terms is needed. Also in p. 35 “The observed rate constants (kobs) for fluorescence decrease (initial phase) and increase (second phase) were determined by fitting the data to single exponential equations...” Does this mean the authors fitted the functions separating two segments of the time courses? If that is the case it may be part of the explanation for the unexpected behavior of Fig 2d and S2b.

Response: As outlined earlier, stopped-flow data has been removed from the revised manuscript or moved to the Supplementary Data. Therefore, the issues referred to here are no longer relevant.

One last question that is just terminology, it says in line 51 “H2O2 itself is a weak oxidant”, not at all, oxidant strength is usually measured by the standard reduction potential and H2O2 is one of the strongest non-radical oxidants in biology, it is slow in many reactions, but not a weak oxidant.

Response: We agree with the reviewer and have changed the wording to “H₂O₂ is a strong but kinetically hindered oxidant” (page 3)

Reviewer #3 (Remarks to the Author):

H2O2 was reported before to relax bovine arteries through PKG activation by both soluble guanylate cyclase (sGC)/cGMP-dependent and cGMP-independent thiol oxidation/subunit dimerization mechanisms. The idea is not novel. H2O2 was known to act as one of endothelium-derived hyperpolarizing factor in small mesenteric arteries in response to acetylcholine. However, the type of vascular cells affected by both inhibitors BCNU and AUR is unclear.

Response: We agree with the reviewer that previous studies by others reported H₂O₂ to relax arteries via non-canonical activation of PGK1 α . More recently, we reported *cis*-WOOH rather than H₂O₂ to activate PGK1 α (*Nature* 2019;566:548) under inflammatory conditions when IDO1 is expressed in vascular endothelial cells. The present manuscript deals with aspects helping to explain the molecular basis that allows this inflammation-driven switch in signalling molecule from H₂O₂ to *cis*-WOOH at the biochemical level. As H₂O₂- and *cis*-WOOH-mediated signalling is dependent on H₂O₂, we hypothesized that such signalling is regulated by the principal regulators of cellular metabolism of H₂O₂, *i.e.*, GPx and Prx.

The purpose of the experiments describing the vascular effects of BCNU and AUR was to assess this hypothesis. We first show that BCNU and AUR cause relaxation in naïve arteries. We interpret this as the two agents increasing the availability of H₂O₂ to engage in oxidative activation of PGK1 α by attenuating the metabolism of the hydroperoxide by GPx and Prx. We then show that such arterial relaxation is inhibited in arteries from LPS-treated animals that express IDO1. This is observed despite the fact that arteries from LPS-treated animals produce increased amounts of H₂O₂ (*Nature* 2019;566:548) and relaxation to reagent H₂O₂ is not altered, indicating that the decrease in BCNU- and AUR-induced relaxation is neither due to a decrease in arterial H₂O₂ nor a difference in endogenous metabolism of H₂O₂ by GPx and Prx. We therefore interpreted the decreased BCNU- and AUR-induced relaxation to indicate a switch from H₂O₂ to *cis*-WOOH as the oxidant responsible for oxidative activation of PGK1 α . Subsequent experiments aimed at examining this possibility resulting from differences in the metabolism of H₂O₂ *versus cis*-WOOH by GPx and Prx.

Other major comments include

1. The information for functional assay of arteries on myograph is insufficient. The concentration-response effects in response to either BCNU or AUR in noradrenaline-pre-constricted vessels are required although the relaxant effect of AUR is rather transient (biphasic). Were the arteries used without intact endothelium? Do both drugs also relax pre-constricted mouse aortas?

Response: Arteries used in our experiments were intact, and the original version of our manuscript reported the threshold for ACh responses for all vessels and treatments on of page 23. We have not tested the effects of AUR on pre-constricted aorta, as the focus of our work is on blood pressure regulating resistance arteries. In response to the reviewer’s query, we have performed concentration response experiments with AUR and BCNU showing a concentration dependency in relaxation in noradrenalin pre-constricted resistance arteries (**Figure 20**). This data is referred to in the text of the revised manuscript (page 6-7) **and is presented as Fig S1a and Fig S1d in the revised manuscript**.

Figure 20. Auranofin and BCNU concentration response curves. Concentration-dependent effects of auranofin (AU) or BCNU or their respective vehicles (DMSO and EtOH) on noradrenaline pre-constricted mesenteric arteries (n=3 each treatment).

2. The additive effect of combined application of BCNU and AUR is required assuming both inhibitors are capable of increasing H₂O₂ generation in the arteries by blocking recycling of oxidized GPx and Prx enzymes, respectively in the arteries.

Response: We thank the reviewer for this suggestion. We performed these experiments and show that there is an additive effect of BCNU and AUR. This further supports the notion that these agents relax arteries mediated via oxidative activation of PGK1 α by slowing the metabolism of H₂O₂ (**Figure 21**). This data has been added to **Figure 1e** of the revised manuscript.

Figure 21. Additive effect of BCNU and auranofin on relaxation of noradrenaline pre-constricted mesenteric arteries. Comparison of relaxation responses from mice treated auranofin (AUR, 300 nM), BCNU (100 μ M) and AUR and BCNU combined (n=7 per treatment), * p<0.05 (one-way ANOVA with Holm-Sidak multiple comparison test).

3. The effect of catalase should be tested on the relaxation induced by BCNU and AUR.

Response: We respectfully point out that the targets of both BCNU and AUR are intra-cellular enzymes, whereas exogenously added catalase will not enter cells and therefore is not expected to affect relaxation induced by BCNU and AUR.

4. H₂O₂ should be used as the positive control in arteries from WT and PKG1 α KO mice.

Response: The impact of replacing Cys42 with Ser on H₂O₂-mediated relaxation of arteries from naïve mice has been reported previously (*Nat Med* 2012;18:286). Arteries from PKG1 α redox dead knock-in (KI) mice show diminished relaxation in response to H₂O₂ (**Figure 22**).

Figure 22. Differential response of WT and KI vessels to oxidant intervention. Dose-dependent relaxation of WT or KI mesenteric vessels to H₂O₂ (left) and representative traces from WT and KI vessels to H₂O₂. Reproduced from *Nat Med* 2012;18:286.

5. Does NO donor also produce less relaxation in arteries from PKG1 α KO mice?

Response: NO donor-induced relaxation has been tested (*Nat Med* 2012;18:286) and is not altered in arteries from PKG1 α redox dead knock-in (KI) compared with wildtype (WT) mice (**Figure 23**).

Figure 23. Comparison of relaxation responses of WT and KI mesenteric vessels to spermine NONOate (SpNONOate), ACh or an EDHF (*i.e.*, acetylcholine in the presence of L-NAME and indomethacin) protocol. Reproduced from *Nat Med* 2012;18:286.

6. LPS may induce iNOS expression to reduce vascular tone while Fig. 1f did not show such effect.
Response: Figure 1f shows relaxation responses in BCNU and AUR. As indicated above, we posit that such relaxation reflects changes in the metabolism of endogenous H₂O₂ rather than nitric oxide. In any case, the experiments were performed in the presence of L-NAME which blocks all isoforms of NOS, thereby eliminating any potential contributions by nitric oxide.

7. H₂O₂ may open vascular smooth muscle potassium channel to cause relaxation. The effect of BCNU and AUR should be tested in arteries constricted by elevated extracellular potassium ions for indication of the role of H₂O₂.

Response: We agree with the reviewer that H₂O₂ may cause arterial relaxation via potassium channels in addition to activating PGK1 α . We therefore have tested the effect of AUR on arteries preconstructed with potassium, as suggested by the reviewer. We observed significantly lower relaxation of K⁺- versus NE-pre-constricted mesenteric arteries in response to 300 nM AUR (**Figure 24**), as suggested by the Reviewer. This information has been added to the text of the revised manuscript (page 6) and is presented in **Fig S1b** of the revised manuscript

Figure 24. Effects of high KCl solution on auranofin-induced arterial relaxation. Comparison of relaxation responses to 300 nM AUR in NE- and KCl-constricted (120 mM) mesenteric arteries from naïve mice (n=6 per treatment). * $p < 0.05$ (Mann-Whitney).

8. Vascular production of H₂O₂ should be measured after inhibitor treatment.

Response: We respectfully consider this request to be beyond the major scope of the present manuscript.

9. It should be tested whether inhibitor treatment can enhance acetylcholine-induced endothelium-dependent relaxation in small mesenteric arteries and such effect should be reduced by H₂O₂-degrading enzyme such as catalase in the myograph chamber.

Response: We respectfully consider this request to be beyond the major scope of the present manuscript. Portillo-Ledesma *et al.* (*Biochemistry* 2018;57:3416) calculated the steady-state concentration of H₂O₂ in presence of Prx2 as 4 nM, and argued that at [H₂O₂] > 4 nM, oxidized Prx2 (as disulfide or sulfenic acid species) accumulates, thus overcoming its antioxidant function and allowing H₂O₂ to react with other targets. In fact, in our original manuscript (see **Figure 7** above), the concentration of endogenous H₂O₂ in “inflamed” arteries is conceivably much higher than 4 nM by assuming 1 g of tissue has a volume of 1 mL. Having established this, we considered the kinetics of this reaction not to be a focus of the present study.

REVIEWERS' COMMENTS

Reviewer #1 (Remarks to the Author):

With the revision, the authors have been very responsible to my comment and performed the requested additional experiments. These further strengthend the hypothesis of the manuscript. I am pleased with the revision and have to further request. The manuscript is now acetable for publication. Best regards, Ralf Brandes

Reviewer #3 (Remarks to the Author):

The authors have addressed my all comments on vascular functional assay satisfactorily. This reviewer has no further questions.